# Oligodendrocytes regulate presynaptic properties and neurotransmission through BDNF signaling in the mouse brainstem

Miae Jang[1], Elizabeth Gould[1], Jie Xu[1,2], Eun Jung Kim[1], Jun Hee Kim[1]*

[1]The Department of Cellular and Integrative Physiology, University of Texas Health Science Center, San Antonio, United States; [2]Children's Medical Center, The Second Xiangya Hospital, Central South University, Changsha, China

**Abstract** Neuron–glia communication contributes to the fine-tuning of synaptic functions. Oligodendrocytes near synapses detect and respond to neuronal activity, but their role in synapse development and plasticity remains largely unexplored. We show that oligodendrocytes modulate neurotransmitter release at presynaptic terminals through secretion of brain-derived neurotrophic factor (BDNF). Oligodendrocyte-derived BDNF functions via presynaptic tropomyosin receptor kinase B (TrkB) to ensure fast, reliable neurotransmitter release and auditory transmission in the developing brain. In auditory brainstem slices from $Bdnf^{+/-}$ mice, reduction in endogenous BDNF significantly decreased vesicular glutamate release by reducing the readily releasable pool of glutamate vesicles, without altering presynaptic $Ca^{2+}$ channel activation or release probability. Using conditional knockout mice, cell-specific ablation of BDNF in oligodendrocytes largely recapitulated this effect, which was recovered by BDNF or TrkB agonist application. This study highlights a novel function for oligodendrocytes in synaptic transmission and their potential role in the activity-dependent refinement of presynaptic properties.

DOI: https://doi.org/10.7554/eLife.42156.001

*For correspondence: kimjh@uthscsa.edu

Competing interests: The authors declare that no competing interests exist.

## Introduction

The formation of complex neuronal networks requires the experience-dependent establishment and remodeling of synapses. The precise control of synaptic function depends not only on neurons, but also on glial cells. Immature oligodendrocytes, located near synapses, make functional synapses with neurons, express neurotransmitter receptors (*Bergles et al., 2000*; *Lin and Bergles, 2004*; *Berret et al., 2017*), and secrete neurotrophic factors such as BDNF (*Bagayogo and Dreyfus, 2009*). Thus, oligodendrocytes are in a prime position to participate in bi-directional communication. In this study, we address whether oligodendrocytes can modulate synaptic function through activity-dependent BDNF signaling.

BDNF regulates neuronal survival and growth in the developing central nervous system (CNS; *Alderson et al., 1990*; *Hohn et al., 1990*; *Rodriguez-Tébar et al., 1990*) and is extensively involved in synaptic transmission and plasticity in various brain regions (*Kang and Schuman, 1995*; *Levine et al., 1995*; *Carmignoto et al., 1997*). For example, activity-dependent BDNF secretion is involved in long-term synaptic plasticity in the hippocampus (*Harward et al., 2016*; *Vignoli et al., 2016*; *Gärtner and Staiger, 2002*). It is known that BDNF signaling regulates vesicular glutamate release at presynaptic terminals (*Pozzo-Miller et al., 1999*; *Tyler and Pozzo-Miller, 2001*), but, because of the low expression of both intracellular and extracellular BDNF in most brain areas, little is known regarding the source and the exact location of action of BDNF at synapses and, specifically, its presynaptic effects.

Glial cells modulate synaptic properties and activities through the secretion of BDNF. Astrocytes recycle BDNF and are involved in the stabilization of long-term synaptic plasticity (*Vignoli et al., 2016*). Microglial BDNF is also an important regulator of synaptic plasticity and function during early brain development (*Parkhurst et al., 2013*). In the CNS, oligodendrocytes express and secrete BDNF, and BDNF secretion is regulated by activation of glutamate receptors (*Bagayogo and Dreyfus, 2009*). We recently showed that immature oligodendrocytes have the capacity to sense neuronal activity and receive glutamatergic inputs in the auditory brainstem (*Berret et al., 2017*), suggesting that neuron-oligodendrocyte communication occurs through chemical signaling. BDNF may thus be a bi-directional signaling factor between oligodendrocytes and neurons.

In this study, we investigated the effects of oligodendroglial BDNF on synaptic functions and the mechanisms whereby oligodendroglial BDNF regulates neurotransmitter release at the presynaptic terminal using mice with reduced BDNF levels ($Bdnf^{+/-}$) and with an oligodendrocyte-specific conditional knockout (cKO) of *Bdnf*. We studied the synaptic functions of oligodendroglial BDNF at the synapse between the calyx of Held terminal and the medial nucleus of the trapezoid body (MNTB) neuron in the auditory brainstem, which is an oligodendrocyte- and synapse-rich brain region. Using immunofluorescence microscopy, electrophysiology, electron microscopy, and in vivo auditory brainstem response (ABR) tests, we found that oligodendroglial BDNF is critical for determining the readily releasable pool (RRP) of glutamate vesicles and actively participates in glutamate release at the calyx terminals. The results suggest that oligodendrocytes are involved in synaptic transmission and plasticity specifically through BDNF signaling in the developing auditory brainstem region.

## Results

### BDNF and glutamatergic synapses in the auditory brainstem

Immunostaining using the neuronal marker MAP2 and the oligodendroglial marker, Olig1, in brainstem sections from wild-type (WT) mice showed that BDNF is highly expressed in MNTB principal neurons (*Figure 1A*). It is of note that oligodendrocytes located close to the calyx–MNTB neuron synapse also expressed BDNF. BDNF expression was notably decreased in all cell types in the MNTB in $Bdnf^{+/-}$ mice at P21 (*Figure 1A*). To examine the effect of endogenous BDNF on fast glutamatergic transmission in the auditory brainstem, we recorded miniature excitatory post-synaptic currents (mEPSCs) from MNTB principal neurons in brainstem slices from P16–20 WT and $Bdnf^{+/-}$ mice (*Figure 1B*). There was no significant difference in the amplitude or kinetics, including rise and decay times, of mEPSCs in WT and $Bdnf^{+/-}$ mice (amplitude: 39.9 ± 2.51 pA, *n* = 13 vs 33.4 ± 2.92 pA, *n* = 12, respectively, p=0.10; rise time: 0.3 ± 0.01 ms, *n* = 13 vs 0.3 ± 0.02 ms, *n* = 12, respectively, p=0.37; decay time: 0.6 ± 0.03 ms, *n* = 13 vs 0.7 ± 0.07 ms, *n* = 12, respectively, p=0.58, *unpaired t-test*; *Figure 1C–E*). In addition, the frequency of mEPSCs was not statistically different (2.7 ± 0.69 Hz, *n* = 8 vs 2.4 ± 0.41 Hz, *n* = 11 in WT and $Bdnf^{+/-}$ mice, respectively; p=0.76, *unpaired t-test*; *Figure 1F*). However, the amplitude of evoked EPSCs (eEPSCs) triggered by afferent fiber stimulation was significantly smaller in $Bdnf^{+/-}$ mice (3.1 ± 0.31 nA, *n* = 17 in $Bdnf^{+/-}$ mice vs 5.9 ± 0.35 nA, *n* = 9 in WT; p<0.0001, *unpaired t-test*; *Figure 1G,H*). To examine the changes in postsynaptic receptor kinetics or in asynchronous release, we analyzed the decay of eEPSCs. The line corresponding to eEPSC decay was well fit as a single exponential with a time constant tau (τ) =1.0 ± 0.07 ms (*n* = 9) in WT and 0.9 ± 0.05 ms (*n* = 17) in $Bdnf^{+/-}$ mice, which were not significantly different (p=0.31, *unpaired t-test*; *Figure 1G,I*). These results suggest that a reduction in endogenous BDNF results in impaired glutamatergic transmission, which is caused by alterations in presynaptic properties rather than postsynaptic components.

To test the effect of reduced BDNF on presynaptic properties, we examined the paired pulse ratio (PPR), which was similar in both groups (0.8 ± 0.02, *n* = 9 in WT and 0.8 ± 0.03, *n* = 17 in $Bdnf^{+/-}$ mice; p=0.93, *unpaired t-test*; *Figure 2A*). This indicates that a reduction in endogenous BDNF does not alter the $Ca^{2+}$-dependent release probability at presynaptic terminals. Next, we examined the short-term depression and the RRP size of available glutamate vesicles at presynaptic terminals in WT and $Bdnf^{+/-}$ mice. During a train of stimuli at 100 Hz (50 pulses), the amplitude of eEPSCs displayed strong depression, falling to ~20% of the initial amplitude near the end of the train in both WT and $Bdnf^{+/-}$ mice (*Figure 2B*). There was no notable difference in short-term depression between WT and $Bdnf^{+/-}$ mice (*n* = 7 vs 12). To predict the RRP size, the release probability ($P_r$),

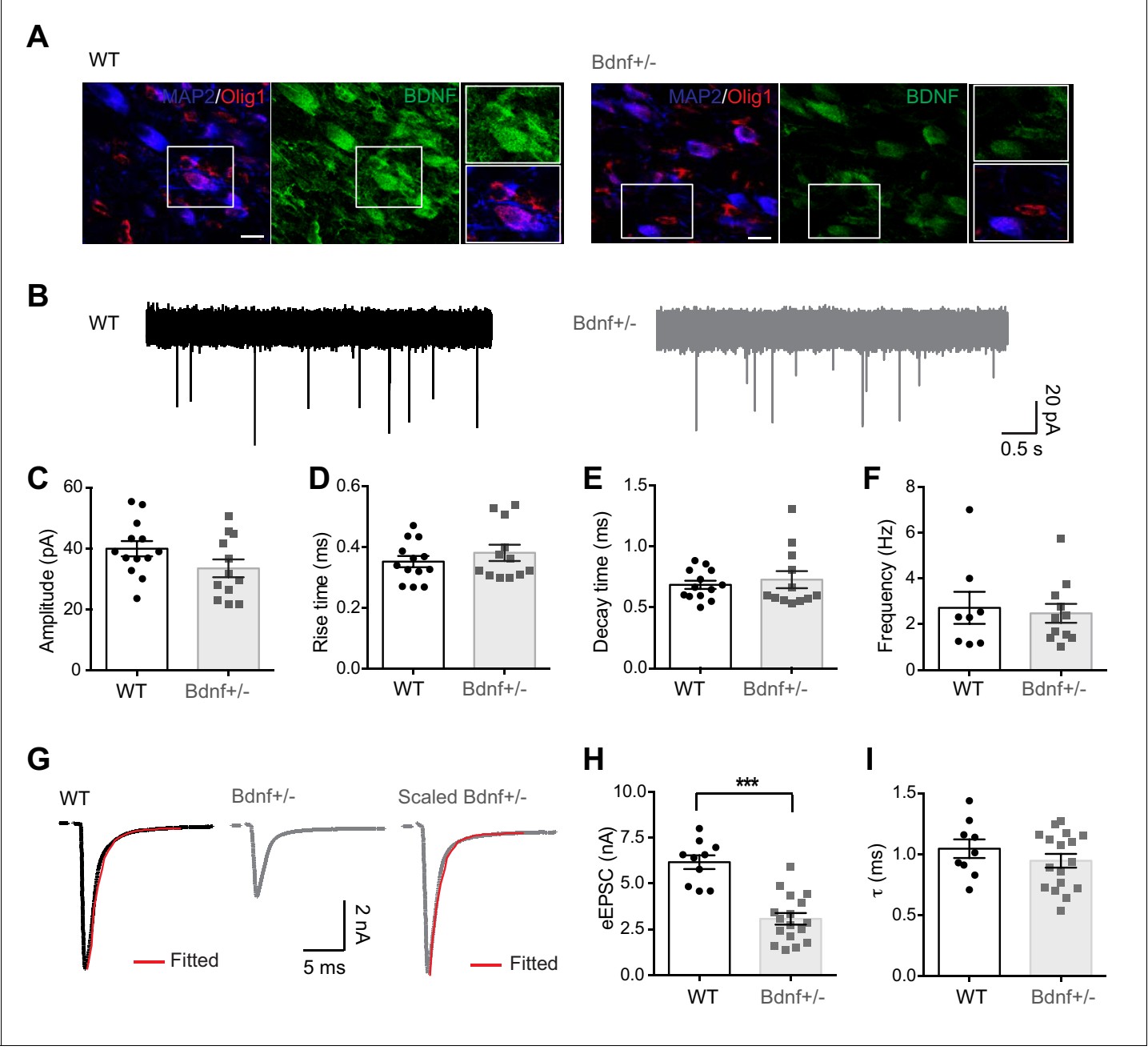

**Figure 1.** Reduction in endogenous BDNF impairs synaptic transmission at the calyx of Held synapse. (A) Representative immunolabeled images for endogenous BDNF expression (green) in the MNTB principal neurons (MAP2, blue) and oligodendrocytes (Olig1, red) in WT and *Bdnf*[+/−] mice at postnatal day (P)21. Images shown are representative of results from *n* = 5 mice per group. Scale bars, 20 µm. (B) Representative traces of mEPSCs from MNTB neurons in WT (black) and *Bdnf*[+/−] mice (gray) at P16–20. (C–F) Quantification of the amplitude (C), rise time (D), decay time (E), and frequency (F) of mEPSCs from WT and *Bdnf*[+/−] mice. (G) A single EPSC evoked by afferent fiber stimulation in WT (black) and *Bdnf*[+/−] (gray) mice. The decay time constant (τ, red) was obtained by single exponential fitting after normalizing the amplitude of EPSCs from *Bdnf*[+/−] mice. (H, I) Summary of the amplitude (H) and decay time constant (I) of eEPSCs from WT and *Bdnf*[+/−] mice. Data are shown as the mean ± s.e.m. ***p<0.001 (unpaired *t*-test).
DOI: https://doi.org/10.7554/eLife.42156.002

and the synaptic vesicle replenishment rate, we used two variants of the cumulative analysis of EPSC trains (*Figure 2C,D*). In the Elmqvist and Quastel (EQ) method (*Elmqvist and Quastel, 1965*), the RRP size was estimated by fitting a line to the linear portion of these data (corresponding to the second through the fourth EPSC) and extrapolating to the *x* axis, we measured the total equivalent

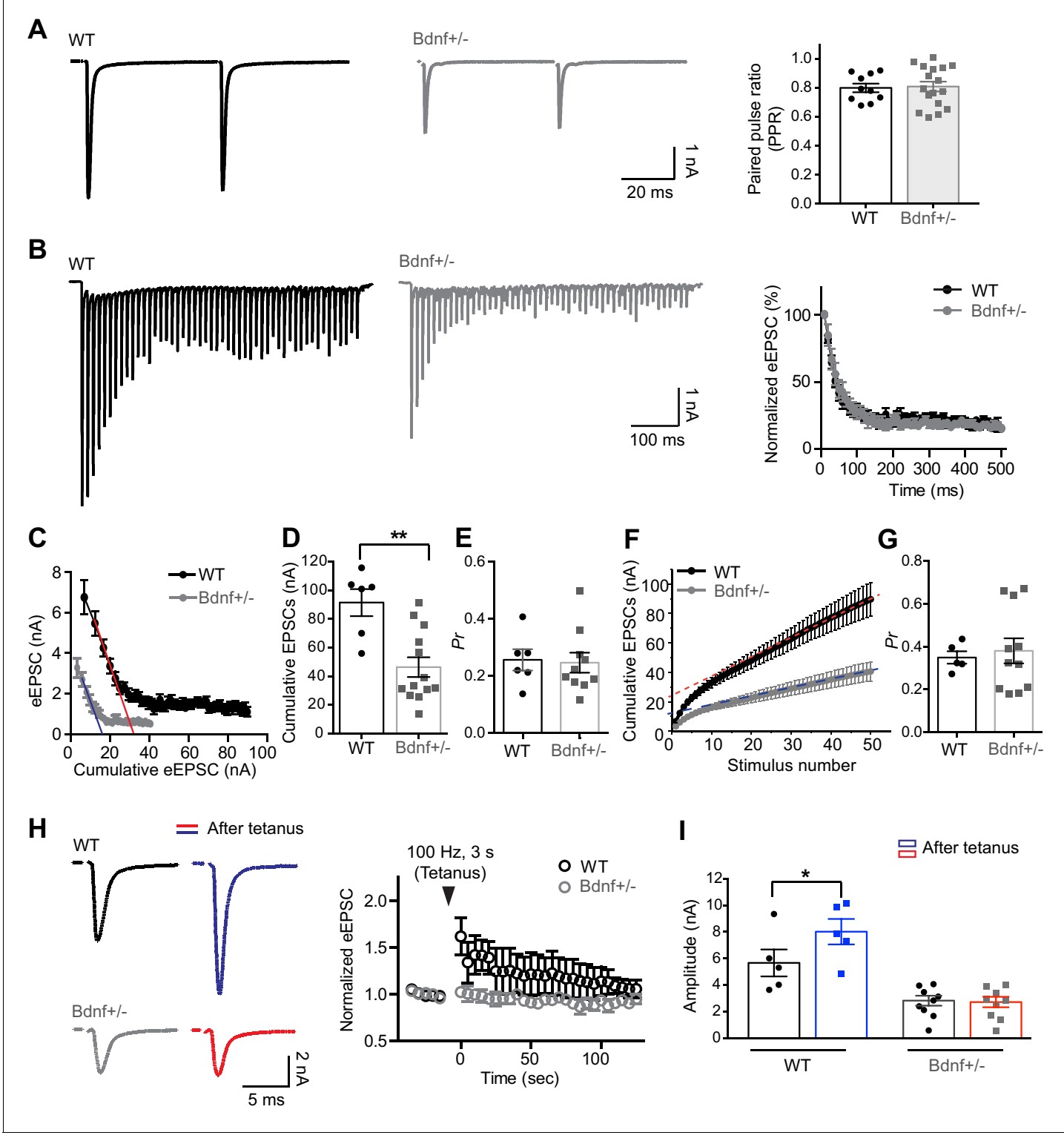

**Figure 2.** Reduction in endogenous BDNF alters presynaptic properties at the calyx terminals. (**A**) Representative traces of EPSCs evoked by paired-pulse stimulation from WT (black) and *Bdnf*[+/−] (gray) mice (at P16, left). Summary of the PPR (right). (**B**) Trains of eEPSCs at 100 Hz stimulation in WT (black) and *Bdnf*[+/−] (gray) mice (left). Normalized amplitude of eEPSCs relative to the first eEPSC amplitude in WT and *Bdnf*[+/−] mice (right). (**C**) Plot of eEPSC amplitudes against the amplitude of the cumulative eEPSC in WT and *Bdnf*[+/−] mice. Plots were linearly fitted from the second through the fourth cumulative eEPSCs (red line for WT and blue line for *Bdnf*[+/−]), which were estimated by back-extrapolated linear fits to the *x* axis to estimate the RRP. (**D, E**) Summary of the cumulative eEPSC size and the release probability (*Pr*) using the EQ method in WT and *Bdnf*[+/−] mice. (**F**) Plot of the cumulative

*Figure 2 continued on next page*

*Figure 2 continued*

eEPSC against stimulus number in WT and $Bdnf^{+/-}$ mice. A line fit to the steady-state points is back-extrapolated to the y-axis to estimate the RRP. (**G**) Summary of the release probability ($P_r$) using the SMN method in WT and $Bdnf^{+/-}$ mice. (**H**) Left, EPSCs evoked at 30 s before and after tetanic stimulation (100 Hz, 3 s) from WT (top) and $Bdnf^{+/-}$ (bottom) mice. Right, plot of normalized eEPSC amplitude after the tetanus relative to the eEPSC amplitude before the tetanus. (**I**) Summary of the amplitude of eEPSCs before and after the tetanus from WT and $Bdnf^{+/-}$ mice. Data are shown as the mean ± s.e.m. *p<0.05; **p<0.01 (unpaired *t*-test; paired *t*-test).

DOI: https://doi.org/10.7554/eLife.42156.003

EPSC at the beginning of the train, which indirectly indicates the available pool of vesicles for release (*Taschenberger et al., 2002*; *Kushmerick et al., 2006*). We plotted the eEPSC amplitudes during a train versus their cumulative amplitudes at the end of a train (*Figure 2C*), which were 91.5 ± 9.46 nA (n = 6) in WT and 46.3 ± 6.89 nA (n = 13) in $Bdnf^{+/-}$ mice (p=0.0016, *unpaired t-test*; *Figure 2D*). The forward extrapolation linear fits revealed that calyces in $Bdnf^{+/-}$ mice had a much smaller RRP of glutamate vesicles as compared with WT mice (17.6 ± 2.73 nA in $Bdnf^{+/-}$ mice, n = 11 vs 33.4 ± 3.31 nA in WT, n = 5; p=0.004, *unpaired t-test*; *Figure 2C*). The RRP divided by the mEPSC amplitude (*Figure 1C*) approximately estimates the number of vesicles, which was reduced in in $Bdnf^{+/-}$ mice (~837 vesicles in WT and ~527 vesicles in $Bdnf^{+/-}$ mice). There was no significant difference in $P_r$, determined as the slop of the linear fit (0.25 ± 0.035 in $Bdnf^{+/-}$ mice, n = 10 vs 0.26 ± 0.037 in WT, n = 6; p=0.8582, *unpaired t-test*; *Figure 2E*).

In the Schneggenburger-Meyer-Neher (SMN) method (*Schneggenburger et al., 1999*), EPSC amplitudes from trains are plotted cumulatively against the stimulus number (*Figure 2F*). A line fit to the steady-state points (the last 10 of 50 points) is back-extrapolated to the y-axis, and the y-intercept divided by the mEPSC amplitude estimates the RRP size. This analysis also revealed that calyces in $Bdnf^{+/-}$ mice had a much smaller RRP of glutamate vesicles as compared with WT mice (9.8 ± 1.29 nA in $Bdnf^{+/-}$ mice, n = 11 vs 19.4 ± 1.71 nA in WT, n = 5; p=0.0008, *unpaired t-test*; *Figure 2F*). Conversely, the release probability ($P_r$), which is calculated by dividing the amplitude of the first eEPSC by the RRP size, was not different in WT and $Bdnf^{+/-}$ mice (0.35 ± 0.02, n = 5 vs 0.38 ± 0.05, n = 11, respectively, p=0.73, *unpaired t-test*; *Figure 2G*). Another interesting finding was the reduced replenishment rate of vesicles in $Bdnf^{+/-}$ mice, which was estimated by the slope of the linear fit (0.61 ± 0.10 in $Bdnf^{+/-}$ mice, n = 11 vs 1.39 ± 0.19 in WT, n = 5; p=0.0018, *unpaired t-test*; *Figure 2F*), indicating that BDNF signaling plays a role in the replenishment of vesicles at the calyx terminal. The values of RRP obtained by the SMN method were smaller than those obtained by the EQ method, because this RRP was measured as the pool decrement during stimulation, whereas the RRP estimated using the EQ Method indicates the size of a pre-existing pool of vesicles (*Neher, 2015*). Taken together, these results show that a reduction in endogenous BDNF decreased the pool of glutamate vesicles available for release at the beginning of a train in $Bdnf^{+/-}$ mice, suggesting that BDNF is important for determining the RRP at presynaptic terminals.

An increase in the RRP of vesicles can contribute to short-term plasticity such as post-tetanic potentiation (PTP; *Habets and Borst, 2005*; *Regehr, 2012*). Thus, we tested whether a decrease in the RRP caused by reduction of BDNF alters PTP at the calyx synapse (*Figure 2H,I*). Tetanic stimulation (100 Hz, 3 s) increased the amplitude of eEPSCs from 5.6 ± 1.01 pA to 8.0 ± 0.96 pA in WT (n = 5; p=0.02, *paired t-test*), indicating a PTP induction, whereas there was no significant increase in the eEPSC amplitude after tetanus in $Bdnf^{+/-}$ mice (2.8 ± 0.37 pA to 2.7 ± 0.39 pA, n = 9; p=0.05, *paired t-test*). Taken together, BDNF controls synaptic plasticity as well as neurotransmitter release by regulating the RRP at the calyx terminal.

## BDNF signaling and exocytosis of vesicular glutamate

We next evaluated the regulatory effect of endogenous BDNF on the exocytosis of vesicular neurotransmitter and $Ca^{2+}$ influx at presynaptic terminals by measuring the membrane capacitance jump ($\Delta C_m$) and voltage-activated $Ca^{2+}$ channel current ($I_{Ca}$) in P9–13 calyces from WT and $Bdnf^{+/-}$ mice (*Figure 3A*). Depolarization induced $Ca^{2+}$ currents and, consequently, $\Delta C_m$ in a pulse duration–dependent manner (2 to 20 ms). $\Delta C_m$ in $Bdnf^{+/-}$ mice was much smaller than that in WT ($\Delta C_m$ after a 20 ms depolarization: 93.4 ± 8.82 fF, n = 11 in $Bdnf^{+/-}$ vs 148 ± 11.32 fF, n = 11 in WT mice; p=0.0013, *unpaired t-test*; *Figure 3A,B*). However, the membrane capacitances of calyces from WT and $Bdnf^{+/-}$ mice were similar (20 ± 1.5 pF, n = 10 vs 21.5 ± 1.35 pF, n = 9, respectively, p=0.43,

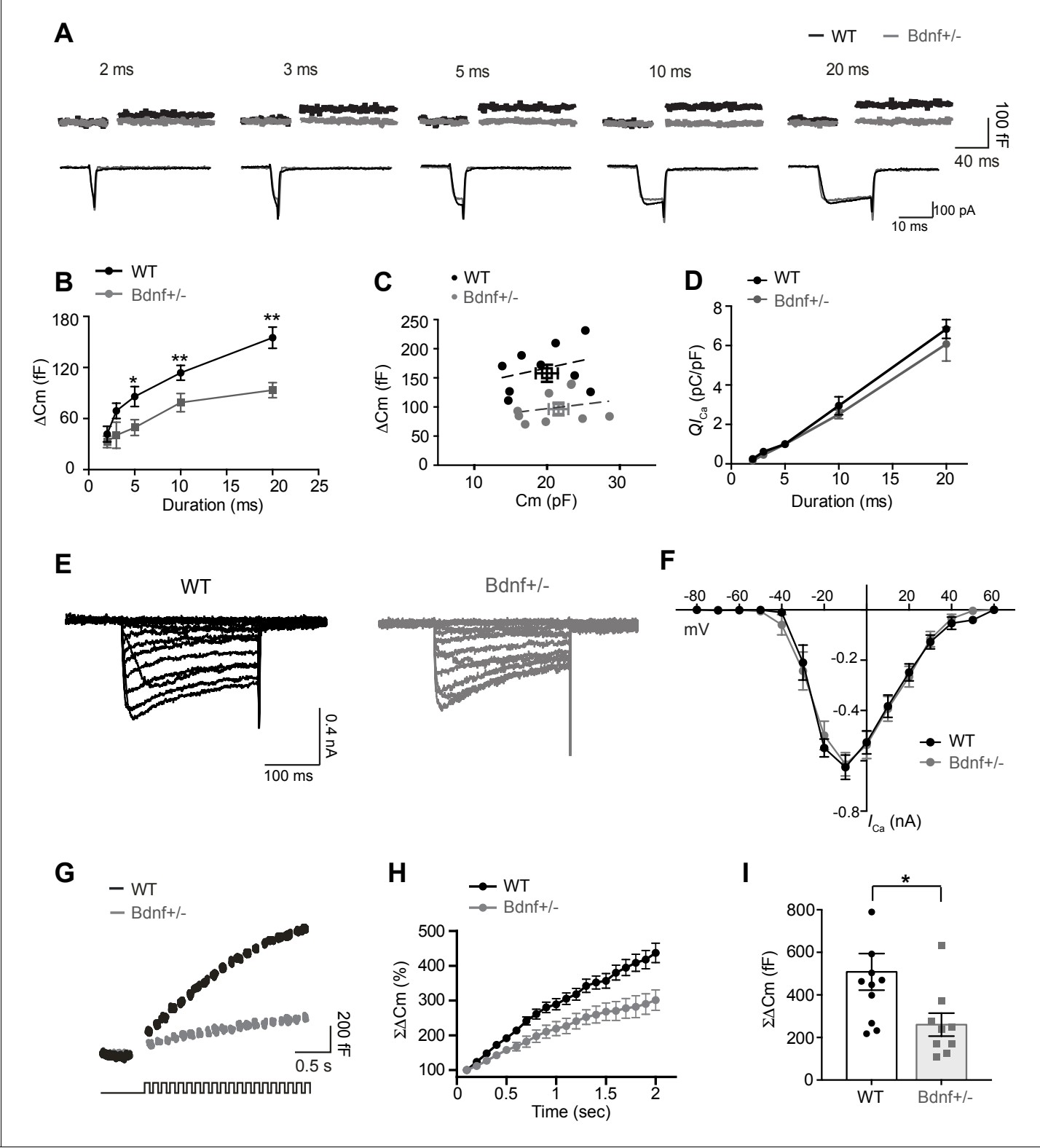

**Figure 3.** Exocytosis of vesicular glutamate is decreased at the calyx terminals in *Bdnf*[+/−] mice. (**A**) Representative traces of membrane capacitance ($C_m$; top) and $Ca^{2+}$ current ($I_{Ca}$; middle) induced by 2-, 3-, 5-, 10-, and 20 ms depolarization (bottom) from P9–13 calyx terminals in WT (black) and *Bdnf*[+/−] (gray) mice. Capacitance within 50 ms after depolarization is not shown to avoid artifacts. (**B**) Summary of capacitance changes ($\Delta C_m$), which are plotted against the depolarization duration in WT and *Bdnf*[+/−] mice. (**C**) Scatter plot: $\Delta C_m$ (elicited by 20 ms depolarization)is plotted against the corresponding resting $C_m$ for each calyx terminal. The squares indicate the mean value. The black and gray lines are linearly fit from each dot for WT and *Bdnf*[+/−] mice, *Figure 3 continued on next page*

*Figure 3 continued*

respectively. (D) Summary of Ca$^{2+}$ current charge ($QI_{Ca}$), which is plotted against the depolarization duration for each genotype. (E) Representative traces of $I_{Ca}$ induced by a 200 ms step-like depolarization (from –80 to 60 mV, $\Delta$10 mV) in WT (black; $n$ = 5) and $Bdnf^{+/-}$ (gray; $n$ = 5) mice. (F) The $I$–$V$ relationship for voltage-activated Ca$^{2+}$ channels at the calyx terminal for each genotype is also shown. (G) Examples of C$_m$ (top) induced by the train of 20 depolarizing pulses (10 ms, 10 Hz; bottom) from –80 to 0 mV in WT (black; $n$ = 11) and $Bdnf^{+/-}$ (gray; $n$ = 9) mice. (H) Summary of the normalized accumulated capacitance jump ($\Sigma\Delta C_m$) relative to the stimulation time for each genotype. Data were normalized relative to the capacitance jump induced by the first 10 ms depolarization. (I) Summary of the $\Sigma\Delta C_m$ after the train of 20 depolarizing pulses (at 2 s) in each genotype. Data are shown as the mean ± s.e.m. *p<0.05; **p<0.01 (unpaired $t$-test).

DOI: https://doi.org/10.7554/eLife.42156.004

The following figure supplement is available for figure 3:

**Figure supplement 1.** Reduction of endogenous BDNF does not affect endocytosis in the calyx of Held.

DOI: https://doi.org/10.7554/eLife.42156.005

unpaired $t$-test; *Figure 3C*), indicating that the sizes of the calyx terminals are similar, and thus the difference in $\Delta C_m$ was not due to the terminal size. In addition, plotting $\Delta C_m$ (for the 20 ms depolarization) as a function of the C$_m$ showed the distribution of C$_m$ as estimated by linear regression analysis ($R^2$ = 0.1 in WT, $n$ = 9; $R^2$ = 0.06 in $Bdnf^{+/-}$ mice, $n$ = 9; *Figure 3C*). Despite their similar resting values for C$_m$, $\Delta C_m$ was much smaller in calyces from $Bdnf^{+/-}$ mice as compared with comparably sized calyces from WT. However, a positive correlation between the resting C$_m$ and $\Delta C_m$ suggest that larger calyces release more vesicles in both genotypes (*Figure 3C*). Next, we examined the effects of BDNF reduction on presynaptic Ca$^{2+}$ channel and $I_{Ca}$ related to changes in exocytosis of glutamate vesicles. There was no significant difference in the presynaptic $I_{Ca}$ charge ($QI_{Ca}$) in response to depolarizing pulses. The smaller $\Delta C_m$ in calyces from $Bdnf^{+/-}$ mice was not attributed to alterations in presynaptic Ca$^{2+}$ currents. In WT calyces, a 20 ms depolarization induced a $I_{Ca}$ of 6.5 ± 0.53 pC/pF ($n$ = 10), whereas in $Bdnf^{+/-}$ calyces, $I_{Ca}$ was 5.9 ± 0.96 pC/pF ($n$ = 9; p=0.50, unpaired $t$-test; *Figure 3D*). In addition, the current–voltage relationship ($I$–$V$) curve for these voltage-activated Ca$^{2+}$ channels at the calyx terminal in WT and $Bdnf^{+/-}$ mice (at P10-12) exhibited a similar pattern with the peak current of −625 ± 48.3 pA vs −614 ± 16.5 pA at −10 mV in WT and $Bdnf^{+/-}$ mice, respectively (n = 15 vs n = 15; *Figure 3E,F*). Furthermore, we examined presynaptic action potential evoked by afferent fiber stimulation, and there was no significant difference in amplitude and half-with of presynaptic action potential, which is associated with Ca$^{2+}$ channel activation and release probability (amplitude, 122.5 ± 7.8 mV in WT, n = 3 vs 124.6 ± 4.4 mV in $Bdnf^{+/-}$, n = 4, p=0.8082 and half-width, 318 ± 41.2 μs in WT, n = 3 vs 283 ± 45.5 μs in $Bdnf^{+/-}$, n = 4, p=0.6123, unpaired $t$-test, data not shown). These data suggest that a reduction in endogenous BDNF decreases exocytosis of vesicular glutamate, but this reduction is not associated with changes in Ca$^{2+}$ influx via Ca$^{2+}$ channels at the presynaptic terminals.

To confirm whether BDNF regulates the RRP and exocytosis of glutamate vesicles at presynaptic terminals, we assessed the vesicle pool size by gradually depleting the RRP using 20 depolarizing pulses (10 ms, 10 Hz). The accumulated capacitance jump ($\Sigma\Delta C_m$), which is a measure of the sum of available glutamate vesicles released by stimulation, was significantly smaller in $Bdnf^{+/-}$ mice (260 ± 54.1 fF, $n$ = 9) relative to WT (508 ± 86.2 fF, $n$ = 11; p=0.04, $t$-test; *Figure 3G–I*). This difference, however, was not associated with the endocytosis rate, which was similar in both groups (19.5 ± 2.75 s, $n$ = 10 in WT vs 17.3 ± 1.91 s, $n$ = 9 in $Bdnf^{+/-}$ mice; p=0.51, unpaired $t$-test; *Figure 3—figure supplement 1*). Therefore, the decreased $\Sigma\Delta C_m$ resulted mainly from the reduction in the RRP size in $Bdnf^{+/-}$ mice. Taken together, these results suggest that the endogenous BDNF level is directly involved in glutamatergic transmission based on its ability to determine the RRP size at the presynaptic terminal.

## BDNF–TrkB signaling and RRP size

We examined whether BDNF function in determining the RRP is mediated by endogenous BDNF at the terminal or by BDNF derived from neighboring cells, which activate TrkB signaling at presynaptic terminals. TrkB was expressed at the calyx terminal and axon, which was immunolabelled with VGluT1 and detected by Alexa 568 dye filling during presynaptic whole-cell recording (*Figure 4A*). We directly activated TrkB using its agonist, 7,8-dihydroxyflavone (7,8-DHF), which binds to the TrkB extracellular domain and activates TrkB-mediated downstream signaling (*Jang et al., 2010*;

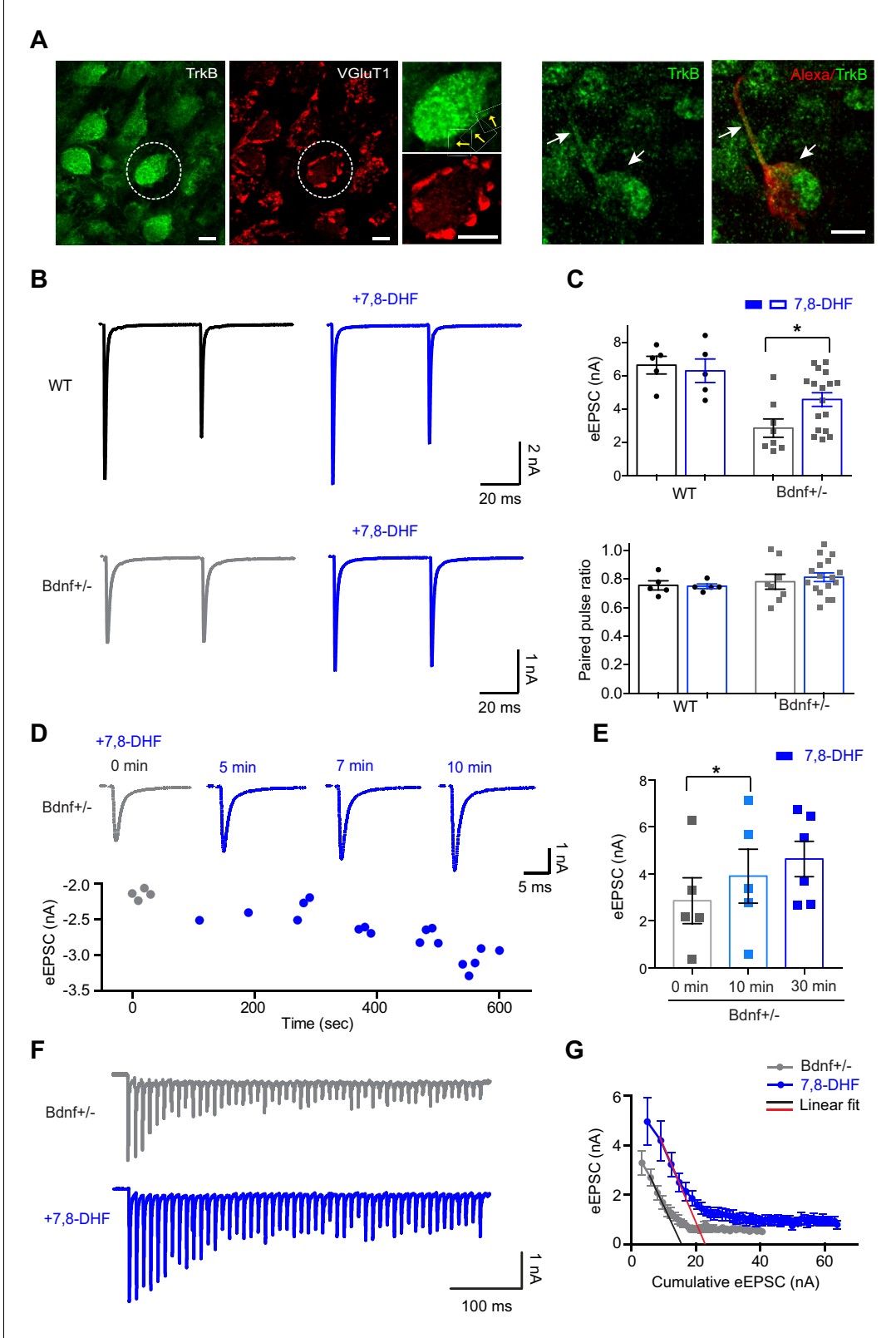

**Figure 4.** The activation of TrkB rescues decreased glutamate release at the calyx terminal. (**A**) Expression of TrkB (green) and VGluT1 (red) at calyx terminals in the MNTB from WT mice (P20). The calyx terminal and axon (arrows, P12 WT mice), filled with Alexa 568 during whole-cell recording, expressed TrkB (green). Scale bars, 10 μm. (**B**) Representative traces of eEPSCs in response to paired-pulse stimulation in the absence or presence of 7,8-DHF for WT ($n$ = 5) and $Bdnf^{+/-}$ ($n$ = 17) mice. (**C**) Summary of the effect of 7,8-DHF on the eEPSC amplitude (top) and the PPR (bottom). (**D**) Top, *Figure 4 continued on next page*

Figure 4 continued
example of eEPSCs at 0, 5, 7, and 10 min after the acute application of 7,8-DHF. Bottom, the amplitude of eEPSCs is plotted against time after 7,8-DHF application. (E) Summary of the acute effect of 7,8-DHF on eEPSCs at different time points. (F) Representative traces of the eEPSC train at 100 Hz from *Bdnf*⁺/⁻ mice in the absence (gray; *n* = 11) or presence of 7,8-DHF (blue; *n* = 7). (G) Plot of eEPSC amplitude against the amplitude of the cumulative eEPSC from *Bdnf*⁺/⁻ mice in the absence (gray) or presence of 7,8-DHF (blue). The black and red lines represent the linear fit from the second through fourth cumulative eEPSCs. Data are shown as the mean ± s.e.m. *p<0.05 (unpaired *t*-test; paired *t*-test).
DOI: https://doi.org/10.7554/eLife.42156.006

*Marongiu et al., 2013*), to determine whether this activation rescues the impaired RRP size and glutamate release at the calyx synapses in *Bdnf*$^{+/-}$ mice. In WT mice, the pre-application of 20 µM 7,8-DHF to brainstem slices (30 min) had no effect on the amplitude (6.6 ± 0.53 nA for control vs 6.3 ± 0.70 nA for 7,8-DHF, *n* = 5, p=0.70, *unpaired t-test*) or PPR (0.75 ± 0.03 for control vs 0.74 ± 0.01 for 7,8-DHF, *n* = 5, p=0.86, *unpaired t-test*) of eEPSCs (*Figure 4B,C*). In *Bdnf*$^{+/-}$ mice, TrkB activation using 7,8-DHF significantly increased the amplitude of eEPSCs (from 2.8 ± 0.54 nA, *n* = 8 to 4.5 ± 0.40 nA, *n* = 17, p=0.02, *unpaired t-test*) without changing the PPR (0.7 ± 0.05 for control, *n* = 8 vs 0.8 ± 0.03 for 7,8-DHF, *n* = 17, p=0.58, *unpaired t-test*; *Figure 4B,C*). Acute application of 7,8-DHF also increased the amplitude of eEPSCs during the 10 min after its application in *Bdnf*$^{+/-}$ mice in a time-dependent manner (2.8 ± 0.97 nA for control, *n* = 5 vs 3.9 ± 1.14 nA for 7,8-DHF at 10 min vs, *n* = 5, p=0.023, *paired t-test*; *Figure 4D,E*). In addition, the pre-application of 7,8-DHF significantly increased the cumulative eEPSC size (from 45.9 ± 7.9 nA, *n* = 11 to 75.2 ± 9.7 nA, *n* = 6; p=0.03, *unpaired t-test*) and partially restored the RRP in *Bdnf*$^{+/-}$ mice (from 17.6 ± 2.73 nA, *n* = 11 to 27.8 ± 3.64 nA, *n* = 7; p=0.03; *unpaired t-test, Figure 4F,G*). These findings suggest that the down-regulation of BDNF–TrkB signaling at the presynaptic terminal impairs the RRP size and glutamatergic transmission in the MNTB.

## Neighboring oligodendrocytes and their BDNF signal

Our results suggested that local BDNF signaling from neighboring cells around presynaptic terminals is critical for determining the RRP at presynaptic terminals during postnatal development. We investigated whether glial cells, specifically oligodendrocytes, are the source of this BDNF signaling. A number of oligodendrocytes are apposed to the calyx synapse in the MNTB during postnatal development (*Figures 1A* and *5A*; *Berret et al., 2017*). To study the specific role of oligodendrocytes in presynaptic functions as BDNF providers, we generated *Bdnf* cKO mice, in which BDNF was specifically deleted in CNPase-expressing oligodendrocytes using the *Cre/loxP* system (*Figure 5B*). To confirm the specificity of the *Cnp*$^{cre}$ line, *Cnp*$^{cre}$ mice were crossed to a GCaMP6f-GFP mouse (or tdTomato reporter) as a reporter line. GFP+ cells were positive for Olig2 and were present next to the calyx synapse, but did not express MAP2, NeuN, and GFAP expression in the MNTB (*Figure 5—figure supplement 1*). This confirms that *Cnp*$^{cre}$ is specific to oligodendrocytes and is not expressed in neurons or astrocytes in the MNTB of the auditory brainstem. In addition, CNP+ cells were positive for CC1, but negative for PDGFRa, indicating that most CNP+ cells in the MNTB are pre-myelinating oligodendrocytes beyond the precursor stage (*Figure 5—figure supplement 1*). *Bdnf* cKO mice (*Cnp*$^{cre}$:*Bdnf*$^{fl/fl}$; *Figure 5B*) were generated by crossing *Cnp*$^{cre}$ mice with mice containing a floxed allele of BDNF (*Bdnf*$^{fl/fl}$). To further confirm the oligodendrocyte-specific depletion of BDNF, oligodendrocytes were isolated via the fluorescent activated cell sorting (FACS) using an O1 antibody, which is specific to oligodendrocytes, or *Cnp*- driven GCaMP6f-GFP (GFP). Utilizing quantitative PCR, we confirmed that the sorted O1+ or GFP+ fraction expressed a substantial level of *Bdnf* in control mice (*Cnp*$^{cre}$:*Bdnf*$^{fl/+}$), whereas the O1+ or GFP+ fraction from *Bdnf* cKO mice showed significantly reduced level of *Bdnf* (*Figure 5—figure supplement 2*). Using presynaptic terminal recordings, we compared *Bdnf* cKO mice with control mice to examine how oligodendroglial BDNF affects presynaptic properties (*Figure 5C*).

The deletion of BDNF in oligodendrocytes significantly decreased exocytosis of glutamate vesicles at the calyx terminal in brainstem slices from *Bdnf* cKO mice. In P9–12 *Bdnf* cKO mice, ΔC$_m$ in response to a 2-, 3-, 5-, 10-, 20-, or 40 ms depolarization was much smaller than in control mice (for 20 ms, 123 ± 20 fF for *Bdnf* cKO, *n* = 19 vs 266 ± 35.4 fF for the control, *n* = 20; p=0.0016, *unpaired t-test, Figure 5D,E*). Longer depolarization induced a larger ΔCm and 40 ms- pulse exhibited saturation of ΔC$_m$ in both control and cKO calyces. ΔC$_m$ resulting from 2 ms depolarization was

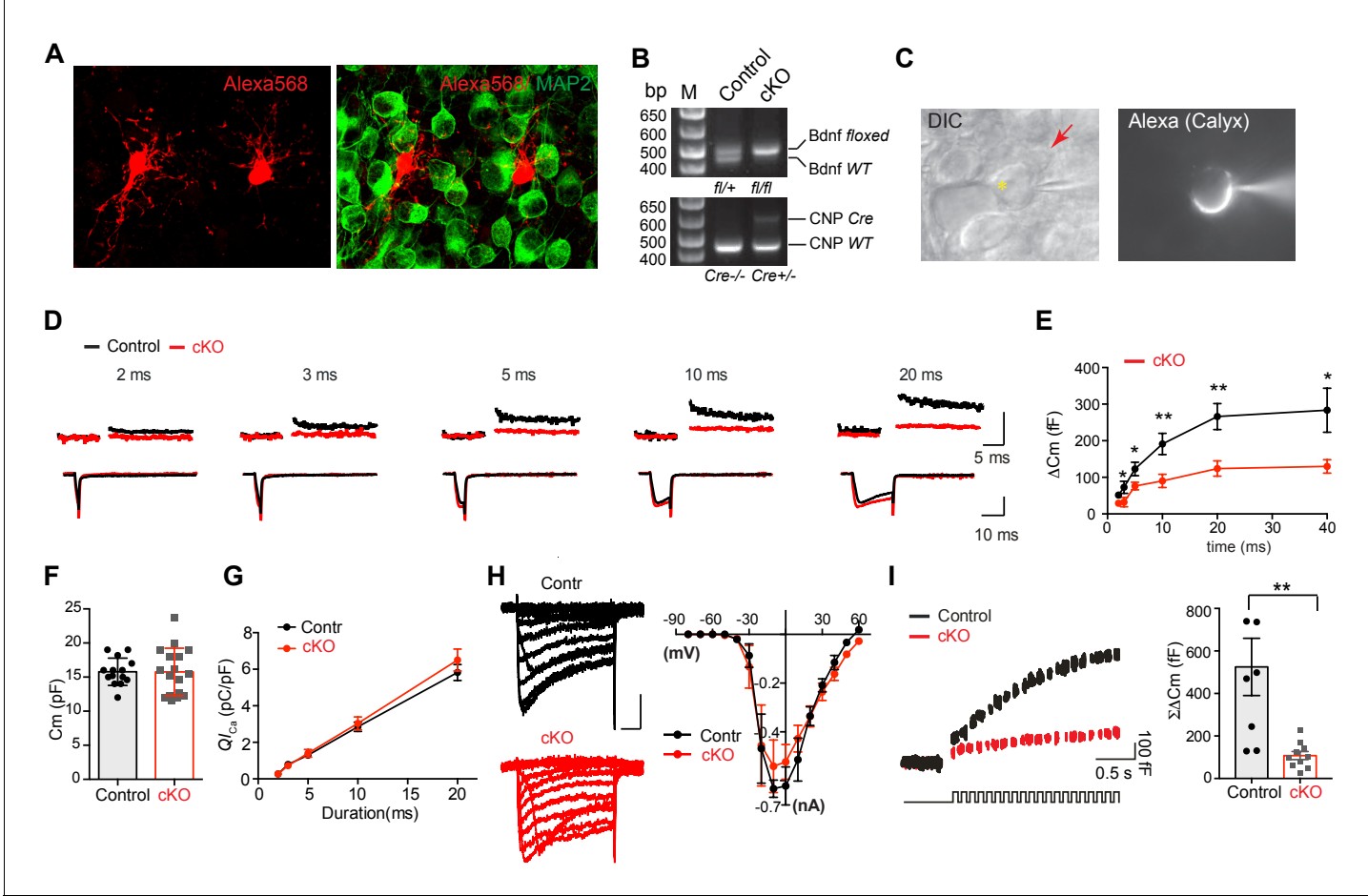

**Figure 5.** Removal of endogenous BDNF from oligodendrocytes affects exocytosis of vesicular glutamate at the presynaptic terminal. (A) Confocal images of oligodendrocytes filled with Alexa 568 using whole-cell recording and MNTB principal neurons, which were immunolabeled with MAP2, from a WT mouse (P10). (B) Conditional deletion of BDNF in oligodendrocytes ($Cnp^{cre}$: $Bdnf^{fl/fl}$). Genotyping PCR using genomic DNA from control and *Bdnf* cKO mice, which are $Cnp^{cre}$: $Bdnf^{fl/+}$ and $Cnp^{cre}$: $Bdnf^{fl/fl}$, respectively. (C) DIC and fluorescence images of the patched calyx terminal filled with Alexa568. Oligodendrocyte (red arrow) was located in close to the calyx synapse in the MNTB. Yellow asterisk indicates MNTB principal neuron. (D) Representative traces for membrane capacitance ($C_m$; top) and $Ca^{2+}$ current ($I_{Ca}$; bottom) induced by 2-, 3-, 5-, 10-, and 20 ms depolarization (bottom) from P10–12 calyx terminals in control (black) and *Bdnf* cKO (red) mice. Scale: 200 fF (top) and 500 pA (bottom), respectively (E) Depolarization duration plotted against $\Delta C_m$ for control (black; 2 ms, $n = 23$; 3 ms, $n = 16$; 5 ms, $n = 23$; 10 ms, $n = 27$; 20 ms, $n = 24$; 40 ms, $n = 3$) and *Bdnf* cKO mice (red; 2 ms, $n = 25$; 3 ms, $n = 10$; 5 ms, $n = 24$; 10 ms, $n = 25$; 20 ms, $n = 23$; 40 ms, $n = 4$). (F) Summary of the resting $C_m$ in WT and *Bdnf* cKO mice. (G) The plot of depolarization duration versus $Ca^{2+}$ current charge ($QI_{Ca}$) was generated from data as in (D) for both genotypes. (H) Left: Representative traces of $I_{Ca}$ induced by a 100 ms step-like depolarization (from –80 to 60 mV, $\Delta 10$ mV) in control (black; $n = 4$) and *Bdnf* cKO mice (red; $n = 5$). Right: The $I$–$V$ relationship for voltage-activated $Ca^{2+}$ channels at the calyx terminal for each genotype is also shown. (I) Summary of the $\Sigma\Delta C_m$ after the train of 20 depolarizing pulses (at 2 s) for each genotype. Data are shown as the mean ± s.e.m. *p<0.05; **p<0.01 (unpaired *t*-test).

DOI: https://doi.org/10.7554/eLife.42156.007

The following figure supplements are available for figure 5:

**Figure supplement 1.** The specificity of the Cnp$^{cre}$ line.

DOI: https://doi.org/10.7554/eLife.42156.008

**Figure supplement 2.** Specific reduction of BDNF in OLs in $Cnp^{cre}$: $Bdnf^{fl/fl}$ mice.

DOI: https://doi.org/10.7554/eLife.42156.009

difficult to resolve in ~50% of *Bdnf* cKO calyces. In both genotypes, the $C_m$ (15 ± 1.9 pF, $n = 14$ for control vs 15 ± 3.5 pF, $n = 17$ for *Bdnf* cKO; *Figure 5F*) and the $Ca^{2+}$ influx during depolarizing pulses were similar (the $QI_{Ca}$ for 20 ms was 5.1 ± 0.45 pC/pF, $n = 13$ for the control vs 5.5 ± 0.39 pC/pF, $n = 11$ for the *Bdnf* cKO; p=0.45, *unpaired t-test*; *Figure 5G*), indicating that the loss of oligodendroglial BDNF impairs vesicular exocytosis without a change in $Ca^{2+}$ channel activation, similar

to what was observed in $Bdnf^{+/-}$ mice. In addition, the current–voltage relationship (*I–V*) curve for these voltage-activated $Ca^{2+}$ channels at the calyx terminal in control and *Bdnf* cKO mice (at P10-12) exhibited a similar pattern with the peak current of $-633 \pm 35.4$ pA vs $-540 \pm 107.1$ pA at $-10$ mV in control and *Bdnf* cKO mice, respectively (n = 4 vs 5; *Figure 5H*). Furthermore, we assessed the $\Sigma\Delta C_m$ during 20 pulses of a 10 ms depolarization at 10 Hz, a protocol that gradually depletes the RRP and thus reflects the RRP size. The $\Sigma\Delta C_m$ evoked by 20 depolarizing pulses was reduced in the *Bdnf* cKO mice ($107 \pm 19.14$ fF, *n* = 10) as compared with that in the control ($525 \pm 134$ fF, *n* = 8; p=0.006, *unpaired t-test*; *Figure 5I*). There was no difference in the endocytosis rate ($19.7 \pm 8.86$ s, *n* = 7 for the control vs $20 \pm 5.28$ s, *n* = 9 for *Bdnf* cKO; p=0.97, *unpaired t-test*; data not shown). These findings suggest that oligodendrocytes are critically involved in determining the presynaptic RRP and vesicular glutamate release through BDNF signaling during postnatal development.

## Role of oligodendroglial BDNF in glutamatergic transmission

We examined the role of oligodendroglial BDNF in glutmatergic transmission in the immature (P10-P12, before hearing onset, *Figure 6A*) and mature calyx synapses (P16-P20, after hearing onset, *Figure 6B*) during postnatal development. The amplitude of eEPSCs was significantly smaller in both immature and mature *Bdnf* cKO mice ($2.4 \pm 0.53$ nA, n = 6 in *Bdnf* cKO vs $4.9 \pm 0.85$ nA, n = 5 in control at P10-12; p=0.032, *unpaired t-test*; *Figure 6A,C*, and $1.6 \pm 0.37$ nA, n = 11 in *Bdnf* cKO vs $6.1 \pm 0.51$ nA, n = 9 in control at P16-20; p<0.0001, *unpaired t-test*; *Figure 6B,E*). In both immature and mature synapses, there was no difference in PPR (*Figure 6D,F*). Next, we examined the RRP size of available glutamate vesicles and its release probability at presynaptic terminals in control and *Bdnf* cKO mice at different ages (P10-12 vs P16-20). Using the EQ method, calyces in *Bdnf* cKO mice had a much smaller RRP of glutamate vesicles as compared with control mice ($9.8 \pm 0.58$ nA in *Bdnf* cKO mice, n = 6 vs $20.1 \pm 2.57$ nA in control at P10-12, n = 3; p=0.0238, *Mann-Whitney test*; *Figure 6G* and $9.6 \pm 3.95$ nA in *Bdnf* cKO mice, n = 6 vs $34.6 \pm 3.39$ nA in control at P16-20, n = 9; p=0.0004, *unpaired t-test*; *Figure 6I*). Conversely, the $P_r$ was not different in both immature and mature control and *Bdnf* cKO mice ($0.33 \pm 0.026$ in *Bdnf* cKO mice, n = 6 vs $0.32 \pm 0.012$ in control at P10-12, n = 3; p=0.7619, *Mann-Whitney test*; *Figure 6H* and $0.41 \pm 0.038$ in *Bdnf* cKO mice, n = 4 vs $0.38 \pm 0.026$ in control at P16-20, n = 9; p=0.4459, *unpaired t-test*; *Figure 6J*). In addition, the SMN method analysis showed a reduction in the RRP and the replenishment rate of RRP in *Bdnf* cKO mice, without significant difference in $P_r$ (*Figure 6—figure supplement 1*). A deletion of BDNF from oligodendrocytes around the calyx synapses significantly impaired the RRP and glutamate release at immature and mature calyx synapses in *Bdnf* cKO mice, suggesting that oligodendroglial BDNF is important for regulating glutamatergic transmission in the auditory brainstem before and after hearing onset.

## Oligodendrocytes and calyx terminal vesicle regulation

To visualize changes in the presynaptic RRP and to quantify the number of glutamate vesicles at the active zone at the calyx terminal, we performed ultrastructural analysis of the calyx–MNTB neuron synapse (at P10-12 and P20) using electron microscopy (EM). Within individual active zones of the calyx terminals, an average of 2–3 docked vesicles was observed in control mice, whereas there were fewer docked vesicles or an absence of docked vesicles at the active zones in the *Bdnf* cKO mice (*Figure 7A*). In immature calyx synapses at P10-12, the number of docked vesicles located within 10 nm from the presynaptic active zone membrane was $2.1 \pm 0.24$ vesicles (62 active zones of three individual cells), whereas in the *Bdnf* cKO mice the number of docked vesicles was significantly reduced to $1.5 \pm 0.18$ vesicles (67 active zones of five individual cells, p=0.0285, *unpaired t-test*, *Figure 7B*, *Figure 7—figure supplement 1*). To test whether oligodendroglial BDNF influences the development of calyces, we assessed the size of calyces using 3D reconstruction of confocal images of the calyx terminals from control and *Bdnf* cKO mice (at P10-12) after presynaptic recordings. The volume of the calyx terminal was not significantly different in *Bdnf* cKO mice ($1378 \pm 143.7$ $\mu m^3$, n = 7 for control and $1199 \pm 146$ $\mu m^3$, n = 6 for *Bdnf* cKO; p=0.3308, *Mann-Whitney test*; *Figure 7—figure supplement 2*). This result was consistent with the membrane capacitance ($C_m$) measurement; there was no difference ($15 \pm 1.9$ pF, n = 14 for control vs $15 \pm 3.5$ pF, n = 17 for *Bdnf* cKO, *Figure 5F*). In mature calyx synapses at P20, $2.7 \pm 0.14$ vesicles were located within 10 nm and $20.2 \pm 0.73$ vesicles were within 200 nm of the active zone of the calyx terminal in the control

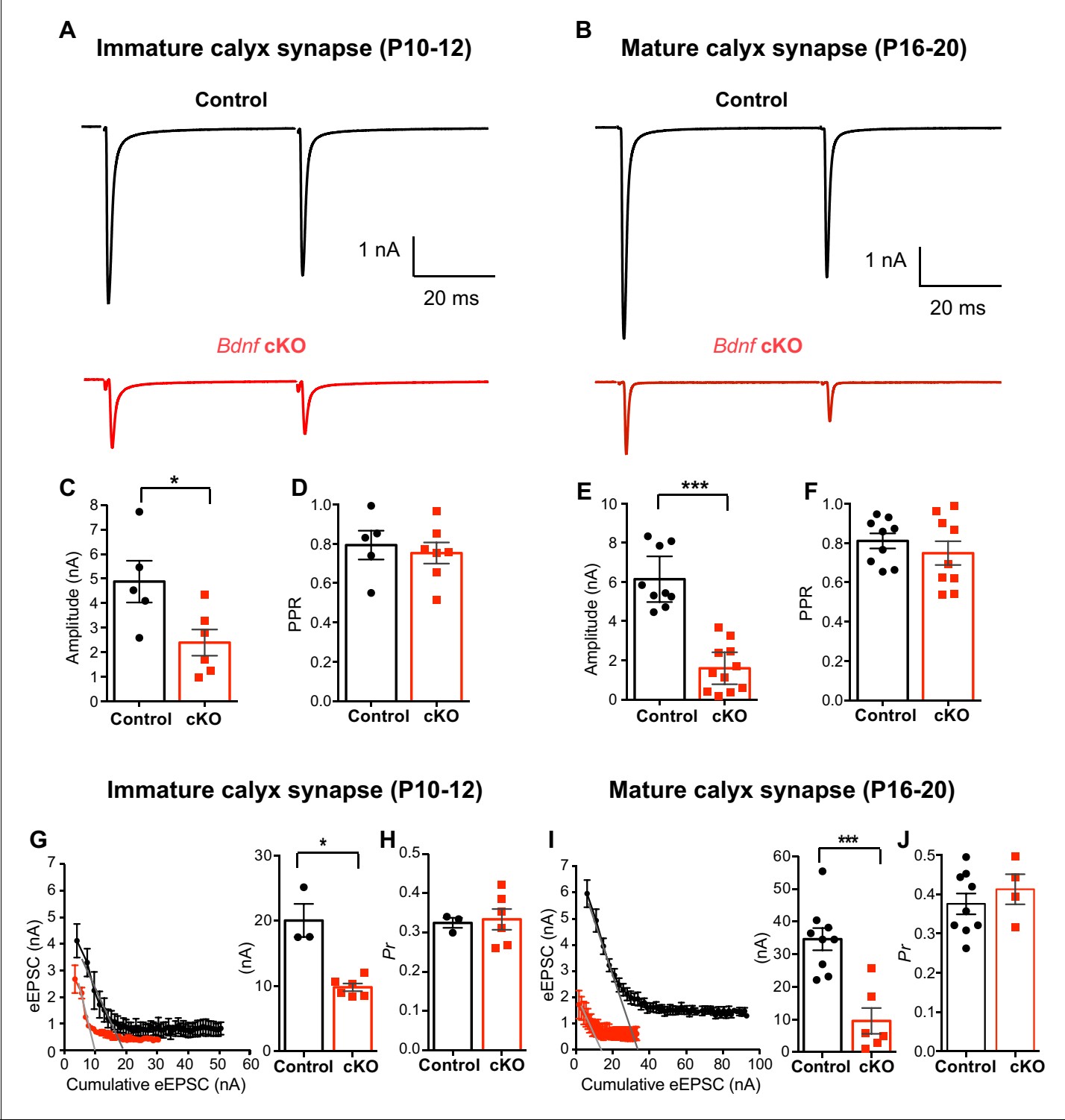

**Figure 6.** Oligodendroglial BDNF critically regulates glutamatergic transmission in the MNTB. (**A, B**) Representative traces of EPSCs evoked by paired-pulse stimulation from immature calyx synapse (at P10-12, **A**) and mature calyx synapse (at P16-20, **B**) in control (black) and *Bdnf* cKO (red) mice. (**C–F**) Summary of the amplitude of EPSCs and the PPR from immature calyx synapses (**C, D**) and mature calyx synapses (**E, F**). (**G–H**) Using the EQ method, plot of eEPSC amplitudes against the amplitude of the cumulative eEPSC in immature calyx synapses from control (black) and *Bdnf* cKO (red) mice. Right: Summary of the RRP size, which was estimated by back-extrapolated linear fits to the x axis. (**G**) Summary of the release probability ($P_r$, **H**). (**I–J**) Summary of the RRP size (**I**) and the $P_r$ (**J**) in mature calyx synapses from control (black) and *Bdnf* cKO (red) mice. Data are shown as the mean ± s.e.m. *p<0.05; ***p<0.001 (unpaired *t*-test; paired *t*-test).

*Figure 6 continued on next page*

*Figure 6 continued*

DOI: https://doi.org/10.7554/eLife.42156.010

The following figure supplement is available for figure 6:

**Figure supplement 1.** RRP and replenishment rate of calyces in cKO.

DOI: https://doi.org/10.7554/eLife.42156.011

(counted in 166 active zones from four cells; *Figure 7A,C*). In *Bdnf* cKO mice, $1.0 \pm 0.12$ and $19.1 \pm 0.75$ vesicles were located within 10 nm and 200 nm of the active zone, respectively, (196 active zones from five individual cells; <10 nm, p<0.0001; <200 nm, p=0.26, *unpaired t-test*; *Figure 7C*). Thus, the number of docked vesicles was significantly decreased in both immature and mature calyces in the *Bdnf* cKO mice. These anatomical changes in presynaptic terminals strongly indicate that oligodendroglial BDNF signaling is important for determining the RRP and specifically for mobilizing glutamate vesicles at the presynaptic terminal during postnatal development.

We next tested whether activation of presynaptic TrkB using an agonist can recover the reduced number of docked vesicles at the active zone in *Bdnf* cKO mice. Auditory brainstem slices from control and *Bdnf* cKO mice (at P20) were prepared for EM imaging after 30 min pre-treatment with 7,8-DHF (20 µM) as described in *Figure 4*. Application of 7,8-DHF recovered the reduced number of docked vesicles to $2.2 \pm 0.22$ within 10 nm of the active zone (63 active zones from three individual cells). There was no change in the number of docked vesicles within 200 nm of the active zone (*Figure 7A,C*). This result indicates that activation of BDNF-TrkB signaling rescues the docking defect or impaired mobilization of vesicles at the active zone, resulting in recovery of the reduced RRP in *Bdnf* cKO mice.

## Oligodendroglial BDNF and presynaptic BDNF–TrkB signaling

We next tested whether extracellular application of BDNF or 7,8-DHF can recover the impaired glutamate vesicle release at presynaptic terminals in *Bdnf* cKO mice. The pre-application of BDNF (100 ng/ml) to brainstem slices for 30 min increased $\Delta C_m$ in response to depolarizing pulses at the calyx terminal from *Bdnf* cKO mice. After 20 ms depolarizing pulses, $\Delta C_m$ was much larger at calyces after BDNF application ($200 \pm 12.72$ fF, $n = 5$) relative to untreated terminals from *Bdnf* cKO mice ($93.8 \pm 23.51$ fF, $n = 13$; p=0.04, *unpaired t-test*; *Figure 8A,B*). There were no corresponding changes in $QI_{Ca}$ in treated and untreated terminals (for 20 ms pulses, $5.1 \pm 0.47$ pC/pF, $n = 13$ vs $5.8 \pm 0.65$ pC/pF, $n = 5$, respectively; p=0.42, *unpaired t-test*; *Figure 8B*). Interestingly, the application of BDNF had no effect on $\Delta C_m$ and $QI_{Ca}$ in control calyces with a normal RRP (*Figure 8—figure supplement 1*). In addition, the direct activation of TrkB also rescued the impaired RRP and glutamate release at the calyx terminal in *Bdnf* cKO mice. After 20 ms depolarization pulses, $\Delta C_m$ was much larger at calyces in the presence of 7,8-DHF as compared with those from *Bdnf* cKO without the 7,8-DHF application ($177.6 \pm 12.72$ fF, $n = 5$ vs $93.8 \pm 23.51$ fF, $n = 13$, respectively; p=0.04, *unpaired t-test*; *Figure 8A,B*). There was no change in the $QI_{Ca}$ in the presence of 7,8-DHF (for 20 ms pulses, $5.1 \pm 0.47$ pC/pF, $n = 13$; p=0.42, *unpaired t-test*; *Figure 8B*). Furthermore, the $\Sigma\Delta C_m$ induced by 20 pulses of 10 ms depolarization at 10 Hz was significantly increased by ~100% in the presence of 7,8-DHF in the *Bdnf* cKO ($107 \pm 19.14$ fF, $n = 10$ in the absence of 7,8-DHF vs $328 \pm 60$ fF, $n = 7$ in the presence of 7,8-DHF; p=0.001, *unpaired t-test*; *Figure 8C–E*). The extracellular BDNF application also partially restored the $\Sigma\Delta C_m$ to $399 \pm 120$ fF in the *Bdnf* cKO ($n = 6$; p=0.008, *unpaired t-test*; *Figure 8E*). Thus, the activation of BDNF–TrkB signaling by the application of BDNF or 7,8-DHF partially recovered the impaired RRP and exocytosis at the presynaptic terminal in the *Bdnf* cKO. These findings suggest that oligodendrocyte-derived BDNF activates presynaptic TrkB signaling, which modulates the RRP and enhances glutamatergic transmission in the MNTB during postnatal development.

## Oligodendroglial BDNF and auditory functions

To assess how loss of oligodendroglial BDNF and subsequent synaptic dysfunction influence auditory functions along the central auditory system, we measured auditory brainstem responses (ABRs), which represent the summed synchronized activity of neurons in the auditory pathway (*Kim et al., 2013*), in control and *Bdnf* cKO mice (P20–25). In both, the ABR waveform consisted of five distinct

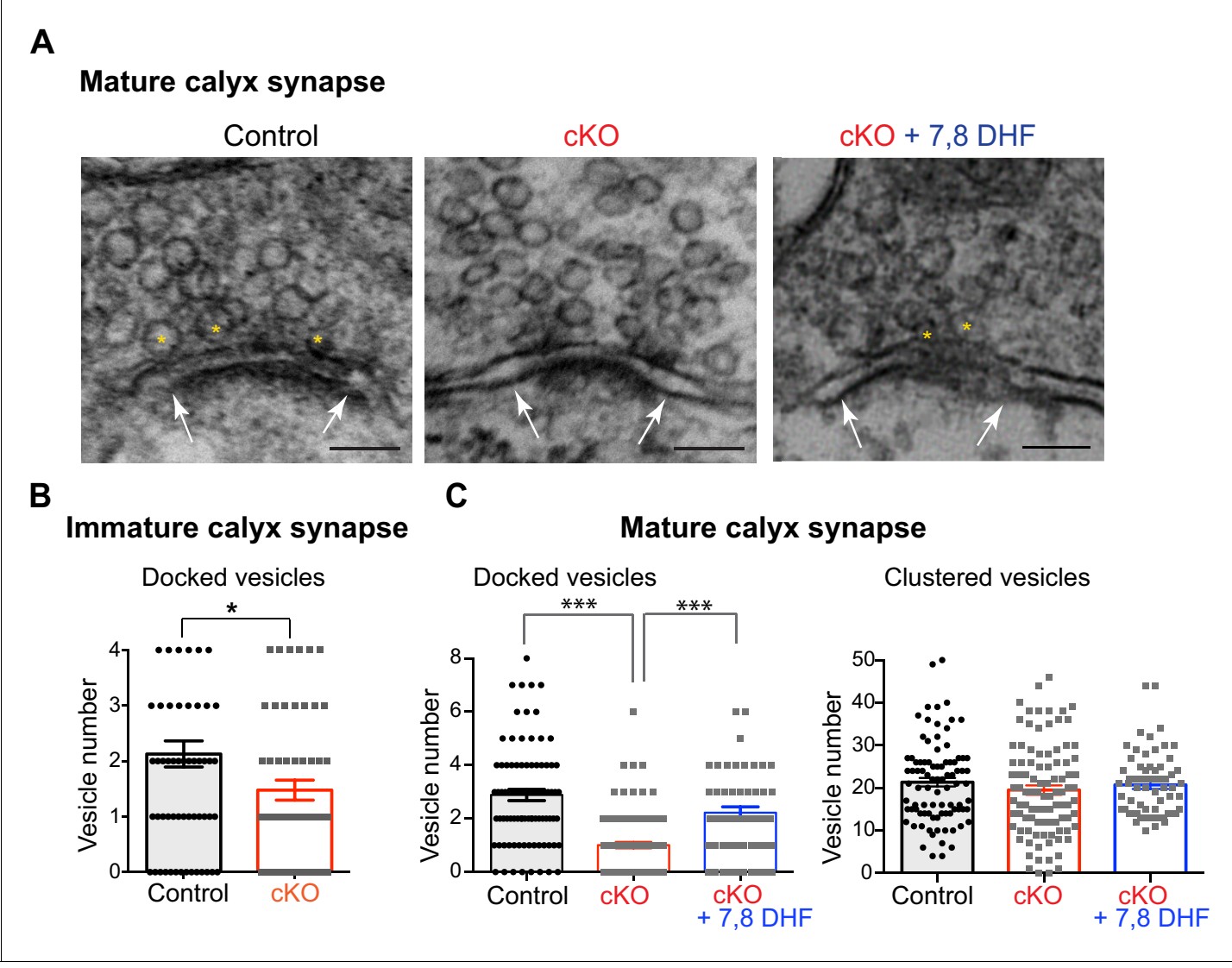

**Figure 7.** Loss of oligodendroglial BDNF reduces the number of docked vesicles at active zones of the calyx terminals. (**A**) EM images of the calyx terminal in the MNTB in control (left) and cKO (middle), and 7,8-DHF treatment on cKO mice (right) at P20. Higher magnification of a presynaptic terminal showed the active zones and synaptic vesicles. The active zones are the dense and dark sites in contact with the MNTB cell membrane (white arrows). Yellow asterisks indicate the docked vesicles within 10 nm of the active zone. The clustered vesicles were located within 200 nm of the active zones. Scale bars, 100 nm. (**B**) Summary of the number of docked vesicles in immature calyx terminals from control (black) and cKO (red) mice at P10. (**C**) Summary of the number of docked vesicles (left) and clustered vesicles (right) at active zones for mature calyx terminals from control (black), cKO (red), and 7,8-DHF treatment on cKO mice (blue) at P20. Data are shown as the mean ± s.e.m. *p<0.05; ***p<0.001 (unpaired *t*-test).
DOI: https://doi.org/10.7554/eLife.42156.012

The following figure supplements are available for figure 7:

**Figure supplement 1.** EM image of the immature calyx terminals in the MNTB in control and cKO mice at P10.
DOI: https://doi.org/10.7554/eLife.42156.013

**Figure supplement 2.** 3D reconstructions of the calyx terminal show reduced terminal volume in Bdnf cKO mice.
DOI: https://doi.org/10.7554/eLife.42156.014

peaks (herein referred to as waves I–V) during the 6 ms following a click stimulus and each wave corresponds to electrical responses from the auditory nerve (wave I) and the ascending auditory pathway (e.g. cochlea nucleus, the superior olivary complex, lateral lemniscus, and inferior colliculus; wave II-V). There was no difference in the threshold of ABRs in response to click stimulation in control and *Bdnf* cKO mice (42.8 ± 2.39 dB vs 42.8 ± 3.04 dB, *n* = 19 vs 14, respectively; *Figure 9A,B*).

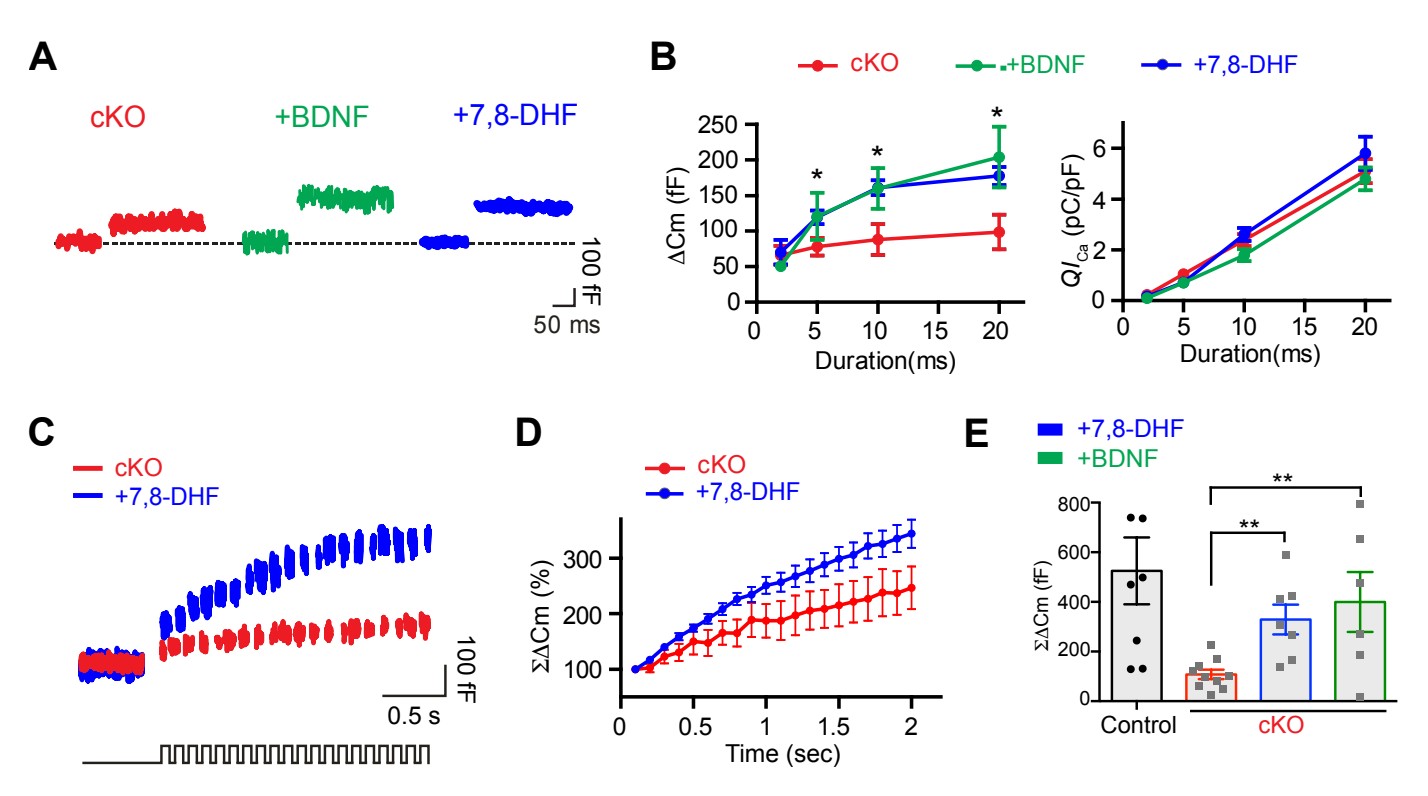

**Figure 8.** Application of extracellular BDNF or 7,8-DHF partially rescues the reduced exocytosis at calyx terminals in *Bdnf* cKO mice. (**A**) Representative traces of $C_m$ (top) and $I_{Ca}$ (middle) induced by 20 ms depolarization from –80 to 0 mV (bottom) at calyx terminals in *Bdnf* cKO mice (P9–13, red) in the presence of BDNF (100 ng/ml; green) or 7,8-DHF (20 μM; blue). (**B**) The duration of depolarizing pulses was plotted versus $\Delta C_m$ (left) and $QI_{Ca}$ (right) for terminals from *Bdnf* cKO slices in the absence (red) and the presence of BDNF (green) or 7,8-DHF (blue). (**C**) Representative traces of $C_m$ (top) induced by the train of 20 depolarizing pulses (10 ms, 10 Hz; bottom) from –80 to 0 mV in terminals from *Bdnf* cKO mice in the absence (red) or presence of 7,8-DHF (blue). (**D**) Summary of the normalized $\Sigma\Delta C_m$ relative to the stimulation time in the absence (red) or presence of 7,8-DHF (blue). (**E**) Summary of $\Sigma\Delta C_m$ of calyx terminals after the train of 20 depolarizing pulses (at 2 s) in the control slices (black) and in *Bdnf* cKO slices in the absence (red) and in the presence of BDNF (green) or 7,8-DHF (blue). Data are shown as the mean ± s.e.m. *p<0.05; **p<0.01 (unpaired *t*-test).

DOI: https://doi.org/10.7554/eLife.42156.015

The following figure supplement is available for figure 8:

**Figure supplement 1.** BDNF application does not affect presynaptic $I_{Ca}$ and exocytosis at the calyx of terminal in control.

DOI: https://doi.org/10.7554/eLife.42156.016

In addition, the latency of wave I, and the time difference between wave I and wave IV, indicating central conduction, did not show significant difference in *Bdnf* cKO mice. We did not observe a significant difference in the amplitude of wave I, whereas the amplitudes of ABR waves II–IV were significantly reduced in *Bdnf* cKO mice (*Figure 9A,B*). In particular, the amplitude of wave III, which reflects the summed neuronal activities of the superior olivary complex, was significantly reduced in the range of click intensities from 55 dB to 85 dB in *Bdnf* cKO mice (at 75 dB, 2.6 ± 0.19 μV, n = 21 for control and 1.7 ± 0.16 μV, n = 16 for the *Bdnf* cKO; p=0.002, *unpaired t-test*; *Figure 9B*). There was no significant difference in the latency of wave I, indicating peripheral conduction, and in central conduction, which was estimated by the time difference between wave IV and wave II (*Figure 9B*). These ABRs indicate that neuronal activity and synaptic synchrony in central auditory nuclei are impaired in *Bdnf* cKO mice. Taken together, the ABRs suggest that endogenous oligodendroglial BDNF regulates the synchrony of synaptic activities and critically influences auditory transmission during postnatal development.

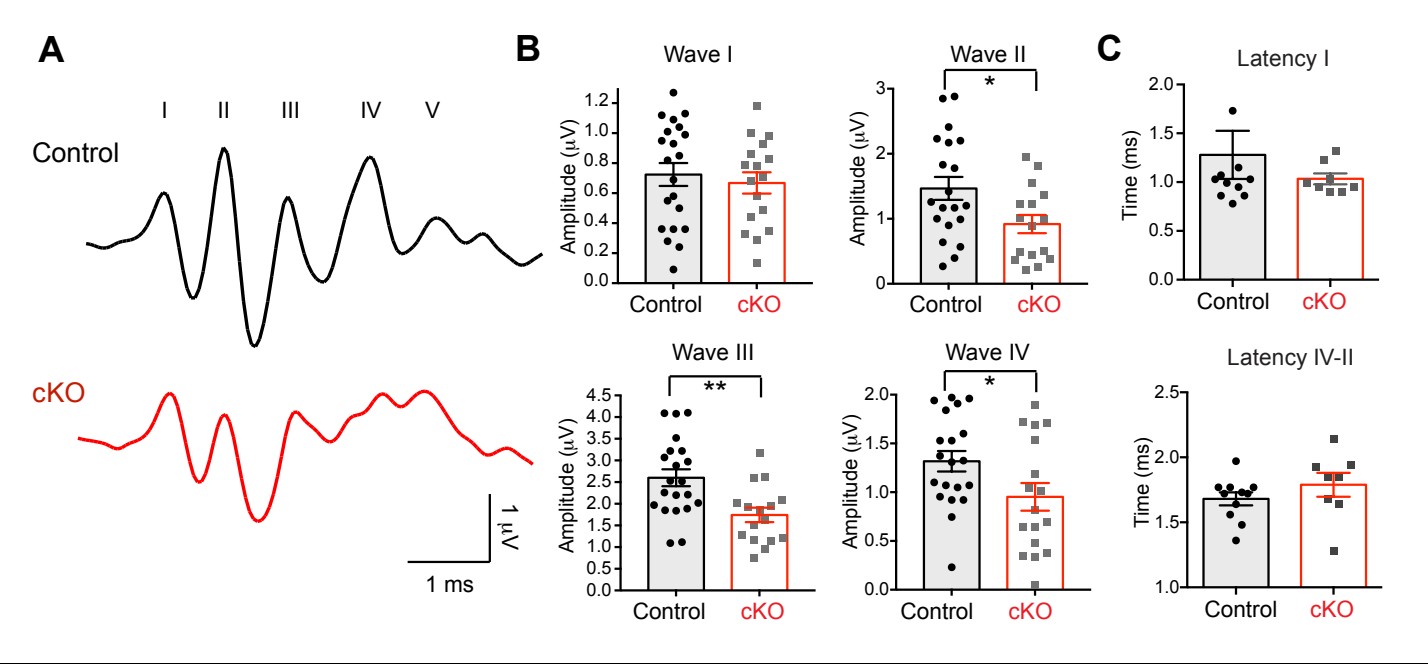

**Figure 9.** The absence of oligodendroglial BDNF impairs the auditory function of *Bdnf* cKO mice. (**A**) Examples of the ABRs in a control (black) and a *Bdnf* cKO mouse (red, both at P25), were recorded in response to a click stimulus of sound (75 dB). Roman numerals indicate peak waves I to V. (**B**) Summary of the amplitude of waves I to IV in response to click stimulus (75 dB), and the latency of wave I and the latency between wave II and IV in control (black) and *Bdnf* cKO mice (red). Data are shown as the mean ± s.e.m. *p<0.05; **p<0.01 (unpaired *t*-test).
DOI: https://doi.org/10.7554/eLife.42156.017

## Discussion

Several forms of cell–cell communication influence synapse formation and pruning. In particular, glial cells are actively involved in synaptic pruning or refinement either during development or in response to brain injury (*Chung and Barres, 2012*; *Karimi-Abdolrezaee and Billakanti, 2012*; *Schafer et al., 2012*). Notably, glial-secreted factors play a critical role in synaptic maturation (*Parkhurst et al., 2013*; *Christopherson et al., 2005*; *Kucukdereli et al., 2011*). BDNF, a neurotrophic factor, is secreted from glial cells and involved in activity-dependent synaptic plasticity (*Zhang and Poo, 2002*; *Lu, 2003*). Oligodendrocytes are considered an important source of BDNF during early postnatal development (*Byravan et al., 1994*; *Dai et al., 2003*). Here, we found that oligodendrocyte-derived BDNF is critical for determining the RRP size and exocytosis of glutamate vesicles at the presynaptic terminal in the developing brainstem.

### BDNF–TrkB signaling at the presynaptic terminal

In this study, we found that a reduction in endogenous BDNF in *Bdnf*[+/−] and *Bdnf* cKO mice impaired glutamatergic transmission without altering presynaptic $Ca^{2+}$ channel activation. The results are comparable to a previous study in the inner ear showing the deletion of endogenous BDNF significantly reduced exocytosis of glutamate vesicles but did not affect $Ca^{2+}$ currents in cochlear hair cells of mice (*Zuccotti et al., 2012*). Furthermore, we found that the application of BDNF or a TrkB agonist (7,8-DHF, 20 μM) led to partial recovery of the reduction in the RRP and in exocytosis of vesicular glutamate in *Bdnf*[+/−] and *Bdnf* cKO, but there was no significant effect in WT or control mice (*Figures 7* and *8*, *Figure 8—figure supplement 1*). However, a previous study in the MNTB of the rat brainstem showed that exogenous BDNF application reduces glutamate release by slowing down presynaptic $Ca^{2+}$ channel activation and inhibiting exocytosis and endocytosis (*Baydyuk et al., 2015*). These conflicting findings may result from the differences between species, ages, or BDNF application method. In particular, the timing of the BDNF signal induction may differentially modulate synapse function as either acute or chronic applications of BDNF can differentially modulate synaptic plasticity (*Sherwood and Lo, 1999*; *Schildt et al., 2013*; *Guo et al., 2018*).

Exogenous administration of BDNF to brain slices has limitations. Depending on administration time, exogenous application could result in the non-specific binding effect of BDNF to presynaptic $Ca^{2+}$ channels, resulting in inhibition of $Ca^{2+}$ channel activation rather than through BDNF-TrkB signaling. Further studies are required to determine the effect of exogenous BDNF on $Ca^{2+}$ channel subtypes expressed in the presynaptic terminal and what aspects of BDNF signaling generate differential responses.

The mechanisms underlying the presynaptic effects of BDNF-TrkB signaling remain elusive. Activation of TrkB leads to the induction of a combination of downstream signaling pathways, including the mitogen-activated protein kinase (MAPK), the PLC pathway, and the phosphatidylinositol 3-kinase (PI3K) pathway, that could modulate synaptic vesicles at the presynaptic terminal (*Yoshii and Constantine-Paton, 2010*; *Reichardt, 2006*). The acute and local effects of oligodendrocyte-derived BDNF on the RRP could be mediated by the increases of intracellular $Ca^{2+}$ levels, which may depend on the activation of the PLC pathway (*Matsumoto et al., 2001*; *Reichardt, 2006*). Recent studies demonstrated that BDNF-induced rise in intracellular $Ca^{2+}$ concentration at the presynaptic terminal was mediated by $Ca^{2+}$ influx through TRPC3 channels, resulting in a transient increase in spontaneous glutamate release (*Cheng et al., 2017*), and/or release of $Ca^{2+}$ from intracellular stores (*Amaral and Pozzo-Miller, 2012*).

## Oligodendrocyte-derived BDNF in the MNTB of the auditory brainstem during early postnatal development

It is important to identify the source of BDNF release at the synapse to understand how BDNF functions and acquires target specificity. BDNF increases by ~10-fold in the mouse CNS in the first 3 postnatal weeks (*Kolbeck et al., 1999*; *Tao et al., 1998*). Although the major source of BDNF in the adult brain appears to be neurons (*Hofer et al., 1990*; *Rauskolb et al., 2010*), BDNF is frequently detected in oligodendrocytes, astrocytes, and microglia in the developing brain (*Dougherty et al., 2000*). BDNF expression is observed in auditory brainstem nuclei in the mouse from P6, and its expression follows the protracted period of development in the auditory pathway, with expression beginning in the ventral cochlear nucleus and continuing to the MNTB and then to the medial superior olive and the lateral superior olive (*Wiechers et al., 1999*; *Hafidi, 1999*). Glial cells may participate in modulating synaptic structure and function during the development of the auditory circuitry by providing a permissive environment through the secretion of BDNF. Oligodendrocytes populate the MNTB prior to astrocytes, indicating oligodendrocytes have a primary role in the maturation of synapses during MNTB development. During the early postnatal weeks, oligodendrocytes are present throughout the auditory brainstem including the MNTB nuclei (as they were at birth), whereas GFAP-positive astrocytes appear in the MNTB during the second postnatal week (*Dinh et al., 2014*). Our immunostaining results, which are consistent with this previous study, showed that there is a greater oligodendrocyte population in the MNTB as compared with GFAP-positive astrocytes by P8 and that most oligodendrocytes are located in proximity to the calyx synapse (data not shown).

This study demonstrated the presence of BDNF in oligodendrocytes in the MNTB of mouse brainstem during early postnatal development. Oligodendrocytes expressed BDNF in the MNTB of the auditory brainstem (*Figure 1A*), and isolated O1+ or $Cnp^{cre}$-driven GCaMP6f-GFP cells expressed a substantial amount of *Bdnf* mRNA (*Figure 5—figure supplement 2*). Previous study demonstrated that oligodendrocytes release BDNF in response to glutamate application (*Bagayogo and Dreyfus, 2009*). Oligodendrocyte processes contact the calyx terminal, which releases glutamate, before forming a myelin sheath during early development (*Figure 5A*). This suggests that glutamate-mediated signaling between oligodendrocytes and the calyx synapses induces BDNF release from oligodendrocytes to increase synaptic strength. Due to the biochemical nature of BDNF, it is thought to act locally at the synapse with limited diffusion within the micrometer range (*Horch and Katz, 2002*; *Sasi et al., 2017*). Oligodendrocytes likely exert a direct impact through the localized secretion of BDNF to the calyx synapses. The results demonstrate oligodendrocytes actively participate in bidirectional neuron–glia communication at the calyx synapse through BDNF-dependent signaling during early postnatal development.

This study utilizes $Cnp^{cre}$ to generate a *Bdnf* deletion specifically in oligodendrocytes. A recent study reported that $Cnp^{cre}$-driven YFP reporter signal was detected in 5.5% of NeuN+ neurons, suggesting a potential limitation on the specificity of recombination of $Cnp^{cre}$ mouse line (*Tognatta et al., 2017*). To address the specificity of $Cnp^{cre}$ in the MNTB, we have analyzed reporter

expression and verified a specific reduction of *Bdnf* in isolated oligodendrocytes. Using two reporter lines, Rosa-GCaMP6f-GFP and Rosa-tdTomato, we identified that <5% of neurons in the MNTB expressed *Cnp*<sup>cre</sup>-driven reporter in early postnatal ages (P10- P20). In addition, using FACS, isolated GCaMP6f-GFP+ cell population contains high levels of Olig2 mRNA with very low levels of Kcc2 mRNA, a neuronal marker (*Figure 5—figure supplement 2*). The majority (>95%) of *Cnp*<sup>cre</sup> expressing cells in the MNTB are oligodendrocytes as shown through immunohistochemistry. We demonstrated that GCaMP6f-GFP+ cells have detectable *Bdnf* mRNA, which was significantly reduced in *Bdnf* cKO mice. There was no significant difference in the level of *Bdnf* mRNA in the GFP− or an O1− fraction, which considered as a non-oligodendroglial population, although there was a trend toward lower *Bdnf* mRNA in the O1− or GFP− fraction. There is the possibility that the small percentage of neurons, affected by *Cnp*<sup>cre</sup>, is sufficient to reduce global levels of *Bdnf* and impact on the synaptic phenotype in the cKO. In cultured neurons, the effect of BDNF within a synapse has been observed to occur within a distance of 4.5 μm (*Horch and Katz, 2002*). Thus, BDNF reduction in a small portion of neurons (<5%) is unlikely to have widespread effects or global impact on the synaptic phenotype observed in the cKO. We interpret that functional alterations of the calyx synapse were caused by the loss of BDNF in oligodendrocytes, which constitute the majority of CNP-expressing cells.

## Bidirectional signaling between oligodendrocytes and nerve terminals

In cultured oligodendrocytes, the activation of glutamate receptors and the phospholipase C pathway enhances the release of dense-core vesicles containing BDNF (*Bagayogo and Dreyfus, 2009*), suggesting that release of BDNF from oligodendrocytes depends on neuronal activity and is mediated by neuron–oligodendrocyte interactions. Our recent study demonstrated that a sub-population of oligodendrocytes interacts with neurons via synapses and displays action potentials in response to intensive neuronal activities in the auditory brainstem (*Berret et al., 2017*). It is intriguing to speculate that bidirectional signaling occurs between oligodendrocytes and nerve terminals at synapses, in which glutamatergic inputs from neurons trigger oligodendrocytes to release BDNF, and then oligodendrocyte-derived BDNF binds to presynaptic TrkB, and finally modulates the glutamate vesicle pool at the nerve terminal. These findings indicate that oligodendrocytes may modulate synaptic plasticity in an activity-dependent manner. An increase in the RRP of vesicles could also contribute to short-term plasticity such as PTP (*Habets and Borst, 2005*; *Regehr, 2012*). We show that the reduction of global BDNF significantly impairs the induction of PTP at the calyx synapse in *Bdnf*<sup>+/−</sup> mice (*Figure 2*). Oligodendrocytes can regulate synaptic strength and plasticity at the calyx synapse by modulating the RRP size through BDNF signaling. Oligodendrocytes that are closely apposed to synapses thus monitor and sense synaptic activity and modulate synaptic plasticity at the presynaptic terminals. In the case of the calyx synapse, this occurs through BDNF–TrkB signaling, which may represent an efficient way for oligodendrocytes to find active nerve terminals and to assist in maintaining synaptic activities. During this critical window of development, when activity-dependent synaptic refinement can occur along the auditory nervous system, oligodendrocytes actively participate in synaptic transmission and plasticity through BDNF signaling in the developing brain.

# Materials and methods

**Key resources table**

| Reagent type (species) or resource | Designation | Source or reference | Identifiers | Additional information |
|---|---|---|---|---|
| Genetic reagent (*M. musculus*) | B6.129S4-Bdnf<sup>tm1Jae</sup>/J | The Jackson Laboratory | stock no.:002266 | PMID:8139657 |
| Genetic reagent (*M. musculus*) | Bdnf<sup>tm3Jae</sup>/J | The Jackson Laboratory | stock no.:004339 | *Rios et al., 2001* |
| Genetic reagent (*M. musculus*) | Cnp<sup>Cre</sup> mice | Dr. Klaus Nave (Max Planck Institute) | MGI no: 3051635 | *Lappe-Siefke et al., 2003* |

*Continued on next page*

*Continued*

| Reagent type (species) or resource | Designation | Source or reference | Identifiers | Additional information |
|---|---|---|---|---|
| Genetic reagent (*M. musculus*) | GCaMP6f | The Jackson Laboratory | stock no.:024105 | Dr. Paukert, UTHSCSA |
| Genetic reagent (*M. musculus*) | tdTomato | The Jackson Laboratory | stock no.:007909 | Dr. Paukert, UTHSCSA |
| Antibody | Mouse monoclonal anti-Olig1 | Millipore | MAB5540 | 1:500 |
| Antibody | Mouse monoclonal anti-MAP2 | Millipore | MAB3418 | 1:200 |
| Antibody | Mouse monoclonal anti-NeuN | Millipore | MAB377 | 1:200 |
| Antibody | Rabbit polyclonal anti-GFAP | DAKO | Z033429 | 1:500 |
| Antibody | Mouse monoclonal anti-CC1 | Millipore | OP80 | 1:200 |
| Antibody | Rabbit monoclonal anti-Olig2 | Abcam | 109186 | 1:100 |
| Antibody | Rat monoclonal anti-PDGFRa | Abcam | AB90967 | 1:300 |
| Antibody | Rabbit polyclonal anti-BDNF | Bioss | BS4989R | 1:100 |
| Antibody | Mouse monoclonal anti-TrkB | Santa Cruz | sc-136990 | 1:50 |
| Antibody | Guinea pig polyclonal anti-VGluT1 | Millipore | AB5905 | 1:1000 |
| Chemical compound, drug | 7,8-Dihydroxyflavone (7,8-DHF) | Sigma | D5446 | 20 μM |
| Chemical compound, drug | BDNF | Millipore | GF301 | 100 ng/ml |
| Chemical compound, drug | TEA-Cl | Sigma | T2265 | 10 mM |
| Chemical compound, drug | 4-AP | Sigma | 275875 | 0.1 mM |
| Chemical compound, drug | TTX | TOCRIS | 1078 | 1 μM |
| Chemical compound, drug | QX314 bromide | TOCRIS | 1014 | 4 mM |
| Chemical compound, drug | Bicuculline | TOCRIS | 130 | 10 μM |
| Chemical compound, drug | Strychnine | Sigma | S8753 | 2 μM |

## Animals

All animal procedures were performed in accordance with the guidelines approved by the University of Texas Health Science Center, San Antonio (UTHSCSA) Institutional Animal Care and Use Committee protocols. BDNF heterozygous ($Bdnf^{+/-}$) mice were generated by crossing $Bdnf^{+/-}$ mice (B6.129S4-Bdnf$^{tm1Jae}$/J; The Jackson Laboratory) with WT mice (C57B[L]6/J). The offspring were genotyped with a standard PCR assay. Primer sequences were as follows: forward, 5'-ATGCGTACC TGACTTTCTCCTTCT-3'; reverse, 5'-ACTGGGTGCTCAGGTACTGGTTGT-3', which amplify a 280 bp and 350 bp fragment for $Bdnf^{+/-}$ mice and a 280 bp fragment for WT.

To create the cKO mice ($Cnp^{cre}$: $Bdnf^{fl/fl}$), mice carrying the floxed allele of $Bdnf$ ($Bdnf^{fl/fl}$; The Jackson Laboratory; *Rios et al., 2001*) were crossed to $Cnp^{cre}$ heterozygous mice (*Lappe-Siefke et al., 2003*). The constitutive KO allele is obtained after Cre-mediated recombination by

crossing $Cnp^{cre}$ mice with $Bdnf^{fl/fl}$ mice to obtain the deletion of $Bdnf$ only in CNPase-expressing cells (**Lappe-Siefke et al., 2003**; **Rios et al., 2001**). Genotypes of all mice were determined by PCR analysis of tail genomic DNA using the appropriate primers: for $Bdnf^{fl/fl}$, forward, 5'-TGTGATTGTG TTTCTGGTGAC-3' and reverse, 5'-GCCTTCATGCAACCGAAGTATG-3', which amplifies a 487 bp ($floxed Bdnf$ allele) and a 437 bp ($Bdnf$ WT allele) fragment; for $Cnp^{cre}$, forward, 5'-GCCACACA TTCCTGCCCAAGCTC-3' and reverse 1, 5'-GTCGCCACGCTGTCTTGGGCTCC-3', and reverse 2, 5'-CTCCCACCGTCAGTACGTGAGAT-3', which amplifies a 400 bp ($Cnp$ WT allele) and 550 bp ($Cnp^{cre}$ allele) fragment. Control mice ($Cnp^{cre}$: $Bdnf^{fl/+}$) were identified by PCR amplification of a 400 bp, 437 bp, and 487 bp fragment, whereas $Bdnf$ cKO mice ($Cnp^{cre}$: $Bdnf^{fl/fl}$) were identified by PCR amplification of a 400 bp, 550 bp, and 487 bp fragment. Recombination efficiency in oligodendrocytes in the $Cnp^{cre}$ mice was determined by transgenic crosses to the GCaMP6f reporter mouse (provided by Dr. Paukert, UTHSCSA or purchased from Jackson Laboratory). All mice were housed in the institutional animal facilities on a 12 hr light/dark cycle. Mice of both sexes aged P8–25 were used for all experiments.

## Slice preparation

Transverse brainstem slices containing the MNTB were prepared from P9–18 mouse pups. After rapid decapitation of the mice, the brains were quickly removed from the skull and immediately immersed in ice-cold low-calcium artificial cerebrospinal fluid (aCSF) containing (in mM): 125 NaCl, 2.5 KCl, 3 $MgCl_2$, 0.1 $CaCl_2$, 25 glucose, 25 $NaHCO_3$, 1.25 $NaH_2PO_4$, 0.4 ascorbic acid, three myoinositol, and 2 Na-pyruvate, pH 7.3–7.4 when bubbled with carbogen (95% $O_2$, 5% $CO_2$; osmolarity of 310–320 mOsm). Then, 200-μm-thick sections were collected using a Vibratome (VT1200S, Leica, Germany). Slices were incubated in a chamber that contained normal aCSF bubbled with carbogen at 35°C for 30 min and then were kept at room temperature. The normal aCSF was the same as the low-calcium aCSF, except that 3 mM $MgCl_2$ and 0.1 mM $CaCl_2$ were replaced with 1 mM $MgCl_2$ and 2 mM $CaCl_2$.

## Electrophysiology

Whole-cell patch-clamp recording was carried out on postsynaptic principal neurons and presynaptic calyx of Held terminals in the MNTB using an EPC-10 amplifier controlled by PATCHMASTER software (HEKA Elektronik, Lambrecht/Pfalz, Germany). Slices were visualized using an infrared differential interference contrast microscope (AxoExaminer, Zeiss, Oberkochen Germany) with a 63 × water immersion objective and a CMOS camera (Hamamatsu Photonics, Hamamatsu, Japan). During experiments, slices were perfused with normal aCSF solution at 2 ml/min at room temperature.

### Presynaptic recording

To measure presynaptic $Ca^{2+}$ currents ($I_{Ca}$) and changes in membrane capacitance ($\Delta C_m$), the borosilicate glass pipettes were filled with a solution containing the following (in mM): 130 Cs-methanesulfonate, 10 CsCl, five sodium phosphocreatine, 10 HEPES, 0.05 BAPTA, 10 TEA-Cl, 4 Mg-ATP, and 0.3 GTP, pH adjusted to 7.3 with CsOH. When filled with the intracellular solution, the pipettes had an open pipette resistance of 4–6 MΩ. Series resistance was <20 MΩ before compensation and <10 MΩ with compensation. Presynaptic $Ca^{2+}$ currents were analyzed after leak subtraction using a 'traditional' p/4 stimulus train in the EPC10-Patchmaster. For identification and morphological analyses, intracellular solutions were supplemented with 50 μM Alexa 568 (Life Technologies, USA). Extracellular aCSF solution contained 10 mM TEA-Cl, 0.1 mM 4-AP, and 1 μM TTX to block $K^+$ and $Na^+$ channels, respectively.

### Postsynaptic recording

For recordings of eEPSCs, the pipettes were filled with a solution containing the following (in mM): 130 Cs-methanesulfonate, 10 CsCl, five sodium phosphocreatine, 10 HEPES, 5 EGTA, 10 TEA-Cl, 4 Mg-ATP, and 0.3 GTP, pH adjusted to 7.3 with CsOH. To this solution, we added 4 mM QX-314 bromide to block the voltage-activated $Na^+$ current. Extracellular aCSF solution contained 10 μM biculculline and 2 μM strychnine to block GABA and glycine receptors, respectively. The holding potential was –70 mV in the voltage-clamp mode. Patch electrodes had resistances of 4–5 MΩ. Series resistance was <20 MΩ, with 80% compensation. Afferent fibers of the calyx of Held synapses

were stimulated with a bipolar electrode (Frederic Haer, Bowdoinham, ME) placed near the midline of the MNTB. An Iso-Flex stimulator driven by a Master 10 pulse at 1.2-fold threshold (<15 V constant voltage) was used. Data were analyzed off-line and displayed with Igor Pro (Wavemetrics, Lake Oswego, OR). Differences were considered statistically significant when p-values were <0.05 by a Student's $t$-test (GraphPad Prism, US). Data are shown as the mean ± s.e.m.

## Immunostaining

Slices used for patch-clamp analysis or fresh brainstem slices (~200 µm thick) were fixed with 4% (w/v) paraformaldehyde in phosphate-buffered saline (PBS) for 20 min. Free-floating slices were blocked in 4% goat serum and 0.3% (w/v) Triton X-100 in PBS for 1 hr and then were incubated with primary antibody overnight at 4°C. The following primary antibodies were used: mouse anti-Olig1 (1:500; Millipore, MAB5540), mouse anti-MAP2 (1:200; Millipore, MAB3418), mouse anti-NeuN (1:200; Millipore, MAB377), rabbit anti-GFAP (1:500; DAKO, Z033429), mouse anti-CC1 (1:200, Millipore, OP80), mouse anti-NeuN (1:600, Millipore, MAB377), anti-Olig2 (1:100, Abcam, 109186), rat anti-PDGFRa (1:300, abcam, AB90967), rabbit anti-BDNF (1:100; Bioss, BS4989R), mouse anti-TrkB (1:50; Santa Cruz, sc-136990), and guinea pig anti-VGluT1 (1:1000; Millipore, AB5905). Tissues were then incubated with different Alexa-conjugated secondary antibodies (1:500; Invitrogen) for 2 hr at room temperature. After three rinses with PBS, slices were coverslipped using mounting medium with 4′,6-diamidino-2-phenylindole (DAPI; Vectashield; Vector Laboratories) to counterstain cell nuclei. Stained slices were viewed on a confocal laser-scanning microscope (Zeiss LSM-510) at 488, 568, and 633 nm using 40 × or 60 × oil immersion objective.

## Transmission EM

Animals were anesthetized and intracardially perfused with normal saline. Brains were removed and 400-µm-thick samples of brainstem MNTB area were dissected out followed by primary fixation in 1% glutaraldehyde/4% paraformaldehyde. Further processing was performed by the UTHSCSA Electron Microscopy Lab. Briefly, each brainstem was post-fixed with 1% Zetterqvist's buffered osmium tetroxide, dehydrated, and embedded in PolyBed resin at 80°C in an oven. Tissue containing the MNTB, which is innervated by calyces of Held, was cut into 90 nm ultrathin sections and placed on copper grids. The sections were then stained with uranyl acetate and Reynold's lead citrate. The samples were imaged on a JEOL 1400 electron microscope using Advanced Microscopy Techniques software. The calyx of Held terminals contacting cell bodies of MNTB principal neurons were recognizable as a cluster of cells located medially in the superior olivary complex (*Taschenberger et al., 2002*). A total of 166–196 synapses were analyzed from five animals for each group (control and *Bdnf* cKO). The number of docked vesicles per the active zone was measured for each synapse at a final magnification of 80,000×. The active zone was defined as the dark presynaptic density contacting the postsynaptic density. Docked and clustered vesicles were defined as those within 10 nm and 200 nm of the presynaptic active zone, respectively (*Sätzler et al., 2002*; *Taschenberger et al., 2002*).

## ABR recordings

ABR recordings were performed as described (*Kim et al., 2013*). Briefly, mice were anesthetized with 4% isoflurane and maintained with 2% isoflurane during recording (1 l/min $O_2$ flow rate). ABR recordings were carried out in a sound attenuation chamber (Med Associates, Albans, VT). Body temperature of mice was maintained at 37°C using a heating pad. Subdermal needle electrodes for recording were placed on the top of the head (active), ipsilateral mastoid (reference), and contralateral mastoid (ground). The electrical potential differences between the vertex and the mastoid electrodes were amplified and filtered (100–5000 Hz), and a recording window of 10 ms starting at the onset of click sound stimulation was distally sampled at 40-µs intervals. Acoustic stimuli were generated by the Auditory Evoked Potentials Workstation (Tucker-Davis Technologies [TDT], Alachua, FL). Closed-field click stimuli were delivered to the left ear using a series of square waves (0.1 ms duration) through TDT Multi-Field Magnetic Speakers placed 10 cm away from the left ear canal. A repetition rate of sound stimuli of 16/s was transmitted through a 10 cm length of plastic tubing (Tygon; 3.2 mm outer diameter). Sound intensities ranged from 90 to 20 dB with 5 dB decrements and responses to 512 sweeps were averaged. The lowest sound intensity that produced a reproducible

waveform was interpreted as the threshold. Free-field pure tone stimuli were taken at frequencies of 8, 12, 16, 24, and 32 kHz at 70 to 20 dB in decrements of 5 dB.

## Fluorescent activated cell sorting (FACS)

*Bdnf* cKO mice (*Cnp$^{Cre}$: Bdnf$^{fl/fl}$*, n = 3) and control mice (*Cnp$^{Cre}$: Bdnf$^{fl/+}$*, n = 2) were used for FACS experiments. *Cnp$^{Cre}$*: Rosa-GCAMP6f-GFP$^{+/-}$ (n = 3) and *Cnp$^{Cre}$* heterozygous controls (n = 3) were used. A cell suspension was generated from the brainstem using enzymatic (papain, 48 U/mL, P1325, Sigma) and mechanical titration. Dissected brainstem was incubated in dissociation media (145 mM NaCl, 5 mM KCl, 20 mM Hepes, 1 mM Na-pyruvate, 2 mM EDTA, pH 7.2) with papain and DNase (10 μM, 10104159001, Sigma) for 20 min. The tissue was spun down at 300 x g for 5 min and resuspended in 1 ml of dissociation media with DNase (10 μM). Mechanical titration was performed using a 1000 ul pipette tip followed by a 200 ul pipette tip. After dissociation, the suspension was filtered and spun down at 300x g for 7 min. The cells were resuspended in 400 μl of dissociation media. 50 μl were set aside for a no primary antibody control. 1 μl anti-O1 antibody (MAB1327, R and D systems) was added to the remaining 350 μl cell suspension (dilution 1:350) and cells were incubated on ice for 25 min. Cells were washed with 1 ml of dissociation media and spun at 300x g for 7 min. Cells were resuspendend in 400 μl dissociation solution with secondary antibody (Alexa 488 anti-mouse IgM, 1:500) and incubated for 25 min. Cells were washed with 1 ml of dissociation media, spun at 300x g for 7 min, and resuspended in 400 μl of dissociation solution. Sorting was performed on a BD FACSARIA III (BD Biosciences) in the Flow Cytometry Facility at UT Health San Antonio with funding from University and the NIH (NCI P30 CA054174). BD FACS Diva software 8.0.1 was used to visualize forward scatter and side scatter to determine cell population and perform doublet discrimination. 488-labeled O1+ cells were selected, yielding a population of 7,000–25,000 cells. In *Cnp$^{Cre}$*: Rosa-GCAMP6f-GFP$^{+/-}$ mice, GFP+ cells produced a population of 50,000–75,000 cells. Negative cells were also collected with a total of 200,000–1,000,000 cells. Cells were kept on ice prior to RNA isolation.

## Quantitative polymerase chain reaction (qPCR)

RNA isolation was performed using an RNAqueous kit with no modifications to procedure (AM1931, Thermofisher). This kit includes DNase treatment. RNA was quantified using a nanodrop (ND-1000, Thermofisher). Purity was assessed by A260/A280 ratios with values ranging from 1.83 to 1.98. 100 ng of RNA was utilized for each reverse transcription (RT) reaction. RT was performed using Superscript III First-strand synthesis with 1 μl Oligo (dT) primers, 1 μl 10 mM dNTP mix (180180–051, Invitrogen). cDNA was stored at −20C for up to 48 hr. qPCR was performed using the PowerUp SYBR green master mix (A25742, Thermofisher). Primers were used at 0.25 μM. Reactions were manually loaded into optical plates with covers (plates:4309849, covers:4360954, Applied Biosystems). The plate was spun for 30 s. qPCR was performed using 7900HT Fast Real-Time PCR system (Applied Biosystems), data was analyzed using SDS v2.4 (Applied Biosystems) and the determined CT was used for analysis. No outliers were removed. No template controls (NTC) as well as no RT controls did not produce determinable CT values. Gapdh was chosen as a reference. Technical replicates were done in triplicate. Delta CT (mean$_{Gene}$ – mean$_{GAPDH}$) was used to determine Delta Delta CT (Delta CT$_{Pos}$-Delta CT$_{Neg}$) or (Delta CT$_{cko}$-Delta CT$_{CTL}$), relative gene expression was calculated 2$^{(-deltadeltaCT)}$ and normalized to 100%. Mouse primers used follows:

Olig2 Forward: 5'-CAAATCTAATTCACATTCGGAAGGTTG
Olig2 Reverse: 5'-GACGATGGGCGACTAGACACC
Kcc2 Forward: 5'-GGGCAGAGAGTACGATGGC
Kcc2 Reverse: 5'-TGGGGTAGGTTGGTGTAGTTG
BDNF Forward: 5'-TCGTTCCTTTCGAGTTAGCC
BDNF Reverse: 5'-TTGGTAAACGGCACAAAAC
GAPDH Forward: 5'-AGTATGACTCCACTCACGGCAA
GAPDH Reverse: 5'-TCTCGCTCCTGGAAGATGGT

## Statistics

All statistical analyses were performed in GraphPad Prism. For electrophysiology, the n equals the number of individual whole-cell recordings. All electrophysiological experiments were performed in

at least seven independent slices from at least seven individual animals. The in vivo ABR test was performed in at least 20 individual controls and at least 15 individual *Bdnf* cKO mice. Data were analyzed off-line and displayed with Igor Pro (Wavemetrics, Lake Oswego, OR). α values were set to 0.05, and all comparisons were two-tailed. To compare two groups, unpaired t-test or Mann-Whitney U test was carried out. Differences were considered statistically significant when p-values were <0.05 by a Student's *t*-test or Mann-Whitney U test (GraphPad Prism). Data are shown as the mean ± standard error of the mean (s.e.m.)

## Data availability

The authors declare that all data generated or analyzed in this study are available within the article.

## Acknowledgements

We would like to thank Drs. Klaus Nave and Manzoor Bhat for providing the *Cnp^{cre}* mouse line. This work was supported by a grant from the National Institute on Deafness and Other Communication Disorders (NIDCD; R01 DC03157) to J H Kim.

## Additional information

### Funding

| Funder | Grant reference number | Author |
| --- | --- | --- |
| National Institute on Deafness and Other Communication Disorders | R01 DC03157 | Jun Hee Kim |

The funders had no role in study design, data collection and interpretation, or the decision to submit the work for publication.

### Author contributions

Miae Jang, Data curation, Formal analysis, Validation, Investigation, Visualization, Writing—original draft; Elizabeth Gould, Data curation, Validation, Visualization; Jie Xu, Data curation, Formal analysis, Validation, Investigation; Eun Jung Kim, Data curation, Formal analysis, Validation, Investigation, Visualization; Jun Hee Kim, Conceptualization, Data curation, Formal analysis, Supervision, Funding acquisition, Validation, Investigation, Visualization, Methodology, Writing—original draft, Project administration, Writing—review and editing

### Author ORCIDs

Jun Hee Kim  http://orcid.org/0000-0003-0207-8410

### Ethics

Animal experimentation: All animal procedures were performed in accordance with the guidelines approved by the University of Texas Health Science Center, San Antonio (UTHSCSA) Institutional Animal Care and Use Committee protocols (#140045x).

### Decision letter and Author response

Decision letter https://doi.org/10.7554/eLife.42156.020
Author response https://doi.org/10.7554/eLife.42156.021

## Additional files

### Supplementary files

• Transparent reporting form
DOI: https://doi.org/10.7554/eLife.42156.018

## Data availability

All data generated or analysed during this study are included in the manuscript and supporting files.

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
