## [Decision Letter]

[Editors’ note: a previous version of this study was rejected after peer review, but the authors submitted for reconsideration. The first decision letter after peer review is shown below.]

Thank you for submitting your work entitled "Oligodendrocytes regulate presynaptic properties and neurotransmission through BDNF signaling" for consideration by *eLife*. Your article has been reviewed by three peer reviewers, and the evaluation has been overseen by a Reviewing Editor and a Senior Editor. The following individual involved in review of your submission has agreed to reveal their identity: Lu-Yang Wang (Reviewer #2).

Our decision has been reached after consultation between the reviewers. Based on these discussions and the individual reviews below, we regret to inform you that your work will not be considered further for publication in *eLife*.

Summary:

All reviewers commented on the importance of understanding the interactions between oligodendrocytes and axons during development and the skill with which the physiological studies were carried out. The studies are clearly novel and interesting. However, as outlined below, major concerns were raised about the congruence between that ages of the mice subjected to the different manipulations and the veracity of the conclusions based on the analysis, in particular the voltage-dependence of Ca channels and the estimates of RRP. In addition, there is insufficient evidence provided that the genetic manipulations designed to delete BDNF from oligodendrocytes was selective and effective. Overall, these issues would have to be addressed before the study is considered sufficiently impactful to warrant publication in *eLife*.

*Reviewer #1:*

This manuscript from Kim and colleagues addresses an important question in developmental neuroscience – the role that myelinating glia play in shaping the maturation of neuronal excitability and synaptic connectivity. The authors exploit the extraordinary accessibility of the calyx synapse in the auditory brainstem to define how BDNF release from oligodendrocytes influences maturation of these synapses. The studies point to a role of this growth factor in altering the RRP, although the mechanism remains to be determined.

1) The authors generate CNPase-Cre; *bdnf*^fl/fl^;Ai9 (tdTomato reporter) mice to achieve oligodendrocyte selective depletion of BDNF. It is crucial for the interpretation that they demonstrate both the selectivity and efficiency of gene deletion. Figure 5—figure supplement 1 evaluates co-localization between tdTomato and Olig1, which is used as a marker for oligodendrocytes. While there is some co-localization between Olig1 and tdTomato in panel A, many of the Olig1+ cells are not tdTomoto immunoreactive, and there appears to be widespread expression of tdTomato in vascular cells (perhaps pericytes). These panels are labeled as "CNPase", however they appear to show the distribution of tdTomato, as an indication of which cells experienced Cre dependent recombination. It is unusual to use Olig1 as a marker for oligodendrocyte cell bodies, rather than CNPase, CC1, or GSTPi, as Olig1 is expressed by other glial cells. The promiscuous expression of tdTomato in these mice raise concerns about specificity of the recombination, which is an issue, given the widespread expression of BDNF (see Figure 1). In addition, they have not provided evidence that this manipulation resulted in depletion of *bdnf* from oligodendrocytes. The standard experiment would be to isolate oligodendrocytes in this region by FACS and then perform qPCR to show that the mRNA is no longer detected. Immunolabeling for BDNF with the markers described above would unlikely to be sufficient, unless the co-localization were unambiguous.

2) The ABR measurements in Figure 8 indicate that there was a remarkable change in latency to the first peak in the *bdnf* cKO mice. This would suggest that conduction along the auditory nerve is slowed in these animals. This issue should be discussed, in light of the fact that CNPase-Cre will also induce recombination in peripheral Schwann cells.

3) There is no evaluation of whether loss of *bdnf* affects the density of oligodendrocyte progenitors or myelination (or Schwann cells). There are numerous reports that BDNF signaling through TrkB alters progenitor dynamics and the process of myelination. If myelination is altered, it could affect activity propagating along GBC axons and therefore maturation of their terminals. It is possible that there is sufficient BDNF coming from other sources to compensate, but if that is true, why don't these other sources compensate for the loss of oligodendrocyte-derived *bdnf*?

*Reviewer #2:*

By a combination of immunofluorescence labeling, patch-clamp electrophysiology, electron microscopy, and in vivo auditory brainstem response (ABR) tests in transgenic mice with conventional knockout (BDNF^+/-^) or oligodendrocyte-specific conditional knockout of BDNF (cKO), Jang et al. showed that oligodendrocyte derived BDNF is critical for boosting the size of the readily releasable pool (RRP) of synaptic vesicles (SVs) and actively promoting glutamate release from the calyx of Held terminals.

Overall this study presents compelling evidence to support the novel role of BDNF from oligodendrocytes to regulate synaptic transmission by targeting specific quantal parameter with advanced molecular and cellular tools. Conceptually, this study is important for the field of development plasticity in central synapses. This reviewer has no major objections to the main conclusion that the authors have drawn from their data but would like to make a few suggestions for the author to consider for tightening up the loose ends.

1) The authors framed the entire story that BDNF released specifically from oligodendrocytes are critical for modulating the RRP in the context of the critical period of development. However, the choice of age group for different experiments is quite confusing. For example, patch-clamp recordings of EPSCs were from P16-20 mice (mature or nearly mature synapses); immunofluorescence labeling and EM were done in P20-25 mice; and calcium currents and capacitance measurements were done in P9-13 mice (immature synapses). Given that dramatic changes in presynaptic spike waveform, calcium channel-SV coupling distance and morphological remodeling all occur during this period, it is difficult to discern if oligodendrocyte derived BDNF is specifically important for developmental remodeling of immature synapses and/or synaptic signaling in mature synapses. It would impart readers with more confidence if the authors can show the results from the experiments in parallel age groups.

2) With the entirely opposite results from Baydyuk et al., 2015, it is essential to show and validate the expression patterns of BDNF/TrkB for P9-13 age group in order to strengthen the arguments made in this paper against the previous paper.

3) In the capacitance jump experiments, it was stated that there is no difference in the basal membrane capacitance between WT and BDNF(+/-) calyces (Figure 3), but the calyx volume is significantly smaller in morphological analyses (Figure 6). There is an obvious conflict here which needs some explanation.

4) Given the abundant expression of TrkB in MNTB neurons, it is essential to analyze the effects of exogenous ligands on the amplitude and frequency of mEPSCs to ensure the observed effects is purely presynaptic as claimed (Figure 4). It is puzzling in fact that BDNF is not doing anything to postsynaptic neurons with much more abundant TrkB expression to detect BDNF than presynaptic terminals. It is probably too hasty to rule out the role of BDNF in postsynaptic signaling at this stage.

*Reviewer #3:*

The manuscript "Oligodendrocytes regulate presynaptic properties and neurotransmission through BDNF signaling", by Jian et al. examines the role of BDNF signaling on presynaptic function at the p8 -P25 mouse calyx of Held. Using the combination of mouse genetics, patch clamp electrophysiology, ultrastructural analysis, and tests of auditory function the authors make a novel and interesting finding that oligodendrocytes are a critical source of BDNF which is essential for proper synaptic transmission at the calyx of Held synapse. In addition, findings run counter to previous findings that BDNF is a negative regulator of synaptic transmission. In addition, their ability to carry out direct presynaptic recordings and measure calcium currents in combination with capacitance measurements allows for clear mechanistic interpretations of how BDNF regulation of synaptic transmission. Despite the novelty of the findings there are some issues that need to be resolved before acceptable for publication.

1) The mechanism by which BDNF rescues the RRP is unclear. The authors clearly show that with their capacitance measurements there is a reduction in the readily releasable pool at the calyx and that this can be rescued by application of BDNF. In addition, in the absence of BDNF their EM data clearly demonstrates a reduction in docked SV. Therefore, two possible interpretations can be made. Either each calyx contains the same number of active zones (AZs), but in the absence of each AZ has less docked SVs or absence of BDNF during development results in a reduction in the number of AZs. In both cases application of BDNF would result in an increase of docked SVs which would result in an increase on the RRP. The authors should show with EM that BDNF rescues the docking defect seen. In addition, the authors should carry out nonstationary EPSC variance analysis to determine the number of functional AZs in the presence and absence of BDNF.

2) The data concerning BDNF effects on the voltage dependence of activation on Calcium channels is not convincing. These measurements are highly dependent on the pipette series resistance in whole cell voltage clamp. There is no mention of the pipette series resistance during the actual recordings. Based on the example traces presented there appears to be differences in the tail current decay between WT and BDNF^+/-^. This could be due to differences in the quality of the voltage clamp or that reductions in BDNF do impact tail currents. In addition, it is not known if a leak subtraction protocol was performed. The authors should redo their IV analysis with better lower residual patch clamp pipette resistance. Furthermore, they should perform the IV analysis in 1.0 mM external Ca^2+^ to also offset potential issues with voltage clamp quality.

3) The authors demonstrate a reduction in AP evoked release and RRP as defined trains of APs at 100 Hz. It has been well established at the age groups p9-11 that a 3ms step depolarization corresponds to the RRP that can be released by action potentials. In Figure 5 which utilized p9-13, there appears to be no difference between the amount of exocytosis with a 2ms step pulse. Therefore, it is unclear why there is a dramatic effect in the AP RRP measurement and RRP as measured by the step depolarizations. It would be helpful if the authors did a 3ms Cm measurement to directly compare. Additionally, the authors should compare data collected at P9-11 vs P12-13 as mice begin to hear at P12. To accurate determine effects on BDNF regulation of the calyx RRP paired recordings at P9-11 would be ideal but not critical to their interpretations. Finally, why did they not carry out fiber stimulation at this age group too?

4) The calyx of Held goes through structural changes after hearing onset. The volumetric reconstruction of the P20-25 age group revealed significant reductions in calyx volume. However, the Capacitance measurements revealed no change in calyx volume as measured as pF. However, this comes from a different age group compared to the group in volume reconstructions. I am more inclined to trust the volumetric reconstructions since it is very difficult to accurately determine the calyx size with the patch clamp experiments. The authors should carry out volumetric reconstructions as the same age group or measure the calyx volume via a capacitance measurement in the P20-25 age group. Otherwise it is difficult to know why there is such a difference between the two measurements of calyx volume.

[Editors’ note: what now follows is the decision letter after the authors submitted for further consideration.]

Thank you for resubmitting your work entitled "Oligodendrocytes regulate presynaptic properties and neurotransmission through BDNF signalling" for further consideration at *eLife*. Your article has been reviewed by three peer reviewers, and the evaluation has been overseen by a Reviewing Editor and Gary Westbrook as the Senior Editor.

The manuscript has been improved but there are some remaining issues that need to be addressed before acceptance, as outlined below. In particular, there are remaining concerns about the specificity of recombination of the CNPase-Cre mice based on the information provided in the manuscript.

*Reviewer #1:*

Jang and colleagues have made substantial revisions to the manuscript and performed some very nice additional experiments to address concerns raised in the prior review. However, the additional data provided to address concerns about the specificity of recombination in CNPase mice are not sufficient.

1) In Figure 5—figure supplement 1, the authors show immunostaining for NeuN, GFAP, Pdgfra, CC1 and GFP to assess which cell types exhibit recombination. In these panels, GFP refers to GCaMP6f in CNPase-Cre x R26-lsl-GCaMP6f mice. There is diffuse GFP immunoreactivity throughout the GFP panels that is unaccounted for and unexpected if expression were restricted to oligodendrocytes. In addition, these images reveal intense regions of GFP immunoreactivity that co-localize with NeuN (cell in the center left of panel A) and co-localize with Pdgfra (panel C). Perhaps most significantly, in the GFP image of panel C there are several long processes and one cell body (center left) that is much larger than one would expect for an oligodendrocyte. For some reason, although this cell looks like a neuron, it also exhibits weak immunoreactivity to CC1. However, the CC1 immunoreactivity in panel C is very unusual, with several long processes visible (this is not normally observed with CC1 – see panel B) and many of the CC1+ cell bodies have unusual morphologies. Since orthogonal projections are not shown in this figure, it is difficult to assess whether labeling patterns show proximity or true co-localization. Thus, these immunostaining data to not provide strong support for the conclusion that CNPase-Cre mice exhibit selective recombination in oligodendrocytes. A minor point, in the figure legend, the mice are referred to as "CNPase-GCaMP6f-GFP mice". Please indicate the exact genotype of the mice for clarity.

2) The authors also perform PCR analysis on isolated cells to determine the specificity of recombination. However, no experimental details are provided for the FACS and PCR analysis shown in Figure 5. There is insufficient information provided that the isolation procedure was able to isolate a representative population of cells from the brainstem (astrocytes, neurons, oligos, etc.). Procedures used to preserve glia often result in widespread death of neurons, so it is important to assess what types of cells are in this pool.

3) The criteria used to determine positive versus negative O1 cells in Figure 5—figure supplement 2A is not indicated. From the second panel in A, it looks like there is a clear break between the blue/magenta cells and the green cells. However only the O1+ cells that exhibited the highest immunoreactivity were selected for the PCR analysis. This analysis should be repeated by including the blue cells, to provide a more representative sample of the population. It is unclear why GFP was not used to isolate cells in this example, as this would provide a more direct assessment of what types of cells experienced cre-dependent recombination. Because there is concern about neuronal recombination, neurons from this region should be isolated and GFP and BDNF expression assessed selectively in this pool.

4) The authors indicate that >27 cycles were necessary to detect BDNF message in the O1+ population. 27 cycles is typically used as a cutoff, with any product observed with more cycles being attributed to noise/non-specific amplification. This suggests that the amount of BDNF in O1+ cells is at the limit, or below the limit, of detection. Although more cycles were required to detect a product in the cKO O1+ cells, confidence in these data are low for this reason.

5) In Figure 5—figure supplement 2D immunostaining is performed in the cKO mice. The panels show that BDNF immunoreactivity is somewhat lower in O1+ cells in the cKO mice. As a constitutive cre was used, it is surprising that BDNF expression was not abolished. It also appears that BDNF immunoreactivity is also lower in neurons in the cKO mice. Thus, these data raise further questions about the specificity of recombination in these mice.

*Reviewer #2:*

The manuscript "Oligodendrocytes regulate presynaptic properties and neurotransmission through BDNF signaling" is a revision in response to previous comments. In this revision, they have performed new experiments and reanalyzed data. They have performed FACS and isolated oligodendrocytes to demonstrate that BDNF mRNA is knocked down specifically in oligodendrocytes in the BDNF cKO animal. They examined the loss of BDNF on calyx MNTB synaptic transmission and morphology at two developmental times point before hearing P10-12 calyces, and mature calyces P16-20. There data demonstrates that loss of BDNF impacts RRP size and release kinetics and presynaptic ultrastructure in a similar manner at both developmental time points. The authors have redone IV experiments and demonstrate that lack of BDNF does not impact calyx Ca^2+^ currents. Finally, they have analyzed RRP pool estimates using SMN plot in addition to their original EQ plots. These new data and analysis in addition to their previous novel findings on allows the authors to draw clear mechanistic conclusions on how oligodendrocytes BDNF signaling regulates presynaptic neurotransmitter release. These data are significant and represent a significant advancement in our understanding of synaptic plasticity. However, there are a few minor points in regards figures and data that the authors should correct and consider to improve the manuscript.

1) In general, for all bar graphs it would be ideal if the authors also provided a dot plot on top of the bar graphs. This would allow the reader to get an idea of the distribution of the date for the experiments.

2) In Figure 2 it would be ideal to also report the release probability measurements as determined by EQ plot measurement. Please report release probability measurements for EQ and SMN in Figure 6.

3) Figure 3 graphs appear to be missing the 3ms data point on all Panel B and Panel D. There are 5 measurements corresponding to the different depolarization times but only 4 data points on these graphs. In addition, in Panel F of the IV data it appears there is a shift in voltage dependence of activation. There is a lot of variance at the -40 mV point but the mean looks different. The authors should make a simple statement to address this potential issue as it could be due to differences in voltage clamp.

4) Figure 5 appears to be missing plotted IV data. In fact, I am confused by these traces. In the figure legend, it states traces showing 100 ms depolarizations from -80 mV to 0 mV. But there is no panel lettering to these traces.

5) Figure 7 is missing representative EM images from the immature calyx synapse. They need to be added.

6) Figure 7—figure supplement 1 is missing. Also, the volumetric data on the P20-25 calyces that was included in the previous version of the manuscript is now missing.

6) Figure 9. It would be helpful to delineate to the reader who are not auditory physiologist what the ABR waves correspond to.

*Reviewer #3:*

This revised version has significantly improved in its quality. The authors have addressed virtually all my criticisms. The action potential waveform is very important for the argument that RRP is specifically altered by BDNF from oligodendrocytes. I would suggest that stats for spike waveform in two age groups are included in the main figure or text.

[Editors' note: further revisions were requested prior to acceptance, as described below.]

Thank you for resubmitting your work entitled "Oligodendrocytes regulate presynaptic properties and neurotransmission through BDNF signalling" for further consideration at *eLife*. Your revised article has been favorably evaluated by Gary Westbrook as the Senior Editor and Dwight Bergles as the Reviewing Editor. The manuscript has been improved, but there are some remaining issues that need to be addressed before acceptance, as outlined below:

The final conclusion of the study is that oligodendrocytes in the brainstem release BDNF during development, which regulates the structural and functional maturation of calyceal synapses in the mouse MNTB. The effects on glutamate release are profound and well described through paired recordings and anatomical studies. The primary limitation of the study is that one mouse line, Cnp-Cre, was used to alter BDNF levels in oligodendrocytes. Because there is concern about the specificity of recombination in this mouse line, the study would be stronger if a subset of the effects were also assessed in a second oligodendrocyte specific Cre line. The authors did not include a reporter in the Cnp-Cre x Bdnfflox/flox cross to allow assessment of BDNF levels in the recombined cells and the selectivity of the recombination, instead relying on the surface antigen O1 for isolation of oligodendrocyte lineage cells for subsequent qPCR analysis. The authors show that samples enriched in oligodendrocyte lineage cells (O1+) have detectable *bdnf* mRNA by qPCR and that this level is reduced in the cKO mice. Although there is a trend towards lower *bdnf* mRNA in the O1- fraction, the value was not significantly lower (n=3). However, they did not specifically measure whether *bdnf* was lower in neurons (the O1- population presumably contains other cells, such as astrocytes which could mask any effects on neuron *bdnf* levels, as levels are typically higher in those cells). In addition, there is no quantification of BDNF protein in oligodendrocytes at this age, and no immuno-localization or in situ hybridization indicating the presence of BDNF protein or *bdnf* mRNA message in oligodendrocytes in this region. The authors find a small proportion of neurons in the brainstem exhibit evidence of recombination in this mouse line, and recombination outside the oligodendrocyte lineage (e.g. Schwann cells, immune cells) in Cnp-Cre mice has been described by others. This is significant, because BDNF levels are much higher in neurons than oligodendrocytes in cortical regions, although to our knowledge this hasn't been explored in the brainstem at this developmental stage. As BDNF is a secreted protein, any reduction in other cells could therefore contribute to the effects observed. At a minimum, the authors need to thoroughly address these limitations in the Discussion.

Additional concerns:

1) All data points should be added to the histograms, so that the readers can assess the distribution of values from individual experiments.

2) It would be helpful if the authors determined the identity of the tdTomato+ cells in Figure 5—figure supplement 1G by immunostaining.

3) There are numerous typographical errors that need to be corrected (e.g. Figure 5—figure supplement 1 – "Nuen"), the labels in some panels are unclear, and some formatting is off (e.g. Figure 5—figure supplement 2D). Throughout the study the authors used GCaMP6F as a reporter to show the specificity of recombination in the CNP-Cre line, but indicate "GFP" or GCaMP6f-GFP in the graphs and panels. This needs to be corrected throughout, for example, by indicating anti-GFP when indicating the antibody (rather than the protein) or GCaMP6F when indicating the protein.

4) The qPCR primer design should be changed so that the primers span across exons to reduce the possibility of genomic signal contamination.

5) The species should be indicated in the title.

---

## [Author Response]

[Editors’ note: the author responses to the first round of peer review follow.]

Summary:All reviewers commented on the importance of understanding the interactions between oligodendrocytes and axons during development and the skill with which the physiological studies were carried out. The studies are clearly novel and interesting. However, as outlined below, major concerns were raised about the congruence between the ages of the mice subjected to the different manipulations.

We added new figures (new Figures 6 and 7 and supplementary figures) to address these concerns. We examine two different age groups from control and *Bdnf* cKO mice: Immature calyx synapses at P10-12 before hearing, and mature calyx synapse at P16-20 after hearing onset. At both time points, we examine presynaptic structural alterations with ultra-structural analysis using EM, and synaptic functions with EPSC recordings using whole-cell patch-clamp from each genotypes. At P10-12, before hearing onset, presynaptic recordings and EPSC recordings clearly demonstrate a reduction in vesicular neurotransmitter release in *Bdnf* cKO mice, and EM analysis shows that the number of docked vesicles is decreased in cKO mice. At P16-20, after hearing onset, EPSC recordings also show a reduction of vesicular neurotransmitter release in cKO mice, and EM analysis reveals that the number of docked vesicles is decreased in cKO mice. The EPSC and EM data are consistent between the two time points. Thus, we demonstrate that cKO continue to show altered synaptic function after hearing onset. These studies consistently show that OL-driven BDNF critically impacts mobilization of glutamate vesicles to the active zone to form the RRP during postnatal development.

And the veracity of the conclusions based on the analysis, in particular the voltage-dependence of Ca channels and the estimates of RRP.

As recommended in the reviewers’ comments, we re-analyze the voltage-dependence of Ca^2+^ channel currents and revise the *I-V* curve in the calyx terminals from WT, *Bdnf*^+/-^, control, and BDNF cKO mice using whole-cell voltage clamp with a lower series resistance (<20 MOhm before compensation and <10 MOhm after compensation) and analyze them after leak subtraction using HEKA EPC10 and Patchmaster. There are no differences in Ca^2+^ influx and voltage-dependence of Ca^2+^ channels between control and *Bdnf* cKO, or between WT and *Bdnf*^+/-^ mice. We added the newly analyzed data and representative traces in Figures 3 and 5.

In addition, we added the results using the Schneggenburger-Meyer-Neher (SMN) method to estimate the RRP size and the replenishment rate of vesicles from WT, *Bdnf*^+/−^, control, and *Bdnf* cKO. New figures and data were added in Figure 2 and Figure 6—figure supplement 1. These additional studies further strengthen the data supporting alterations in the RRP due to the reduction of BDNF and OL-derived BDNF, specifically.

In addition, there is insufficient evidence provided that the genetic manipulations designed to delete BDNF from oligodendrocytes was selective and effective.

We added new data to confirm that Cre-dependent recombination in *Cn^Cre^* mice is specific to OLs. We show immunostaining for CC1, PDGFRα, and Olig2 as OL markers, and verify their colocalization with the Cre-dependent reporter, GCaMP6f (Figure 5—figure supplement 1). In addition, we perform FACS to isolate O1+ OLs and utilize qPCR to show the level of BDNF mRNA in OLs, as reviewer’s suggested. BDNF RNA is detected in the O1+ OLs isolated from control mice. Importantly, BDNF levels in the O1+ fraction of *Bdnf* cKO mice were significantly reduced compared to control mice. There was no difference in the cKO O1- fraction compared to the control O1- fraction. The results indicate that the knockdown of BDNF occurs specifically in OLs in cKO mice. We added this data in Figure 5—figure supplement 2.

Overall, these issues would have to be addressed before the study is considered sufficiently impactful to warrant publication in eLife.Reviewer #1:[…] 1) The authors generate CNPase-Cre; bdnf^fl/fl^;Ai9 (tdTomato reporter) mice to achieve oligodendrocyte selective depletion of BDNF. It is crucial for the interpretation that they demonstrate both the selectivity and efficiency of gene deletion. Figure 5—figure supplement 1 evaluates co-localization between tdTomato and Olig1, which is used as a marker for oligodendrocytes. While there is some co-localization between Olig1 and tdTomato in panel A, many of the Olig1+ cells are not tdTomoto immunoreactive, and there appears to be widespread expression of tdTomato in vascular cells (perhaps pericytes). These panels are labeled as "CNPase", however they appear to show the distribution of tdTomato, as an indication of which cells experienced Cre dependent recombination. It is unusual to use Olig1 as a marker for oligodendrocyte cell bodies, rather than CNPase, CC1, or GSTPi, as Olig1 is expressed by other glial cells. The promiscuous expression of tdTomato in these mice raise concerns about specificity of the recombination, which is an issue, given the widespread expression of BDNF (see Figure 1). In addition, they have not provided evidence that this manipulation resulted in depletion of bdnf from oligodendrocytes. The standard experiment would be to isolate oligodendrocytes in this region by FACS and then perform qPCR to show that the mRNA is no longer detected. Immunolabeling for BDNF with the markers described above would unlikely to be sufficient, unless the co-localization were unambiguous.

As mentioned above, we added new data to confirm that Cre-dependent recombination in *CNPCre* mice is specific to OLs (Figure 5—figure supplement 1). In addition, we perform FACS to isolate O1+ OLs and utilize qPCR to show the level of BDNF mRNA in OLs, as reviewer’s suggested. O1 is a commonly used marker for OLs and recognizes galactocerebroside. O1 recognizes an external epitope that enables FACS with live stained cells. To validate that we successfully isolated OLs, we compared the relative gene expression of Olig2 in the O1- and O1+ fraction from control mice. The O1+ fraction has Olig2 levels that were 6.73x greater than the O1- fraction (O1- = 100.1 ± 5.2, O1+ = 673.3 ± 17.6, p<0.0001, n = 3 mice). BDNF RNA is detected in the O1+ OLs isolated from control mice (mean cycle threshold (CT) = 27.8 ± 0.08, n = 3 mice). Importantly, BDNF levels in the O1+ fraction of *Bdnf* cKO mice are significantly reduced compared to control mice (Control = 33.4 ± 8.4 vs. cKO = 14.67 ± 0.6, p = 0.0054, n = 3 mice/genotype). There is no difference in the cKO O1- fraction compared to the control O1- fraction (Control = 100.3 ± 7.5 vs. cKO = 89.3 ± 11.8, p = 0.47, n = 3 mice/genotype). The results indicate that the knockdown of BDNF occurs specifically in OLs. We added this data in Figure 5—figure supplement 2.

“To study the specific role of oligodendrocytes in presynaptic functions as BDNF providers, we generated *Bdnf* cKO mice, in which BDNF was specifically deleted in CNPase-expressing (*CNPase*^+^) oligodendrocytes using the *Cre/loxP* system (Figure 5B). […] Using presynaptic terminal recordings, we compared *Bdnf* cKO mice with control mice (*Bdnf*^fl/+^
*CNPCre^–/–^*) to examine how oligodendroglial BDNF affects presynaptic properties (Figure 5C).”

2) The ABR measurements in Figure 8 indicate that there was a remarkable change in latency to the first peak in the bdnf cKO mice. This would suggest that conduction along the auditory nerve is slowed in these animals. This issue should be discussed, in light of the fact that CNPase-Cre will also induce recombination in peripheral Schwann cells.

There is no significant difference in the latency of the first ABR wave from *Bdnf* cKO mice, indicating no significant change in peripheral conduction. We also analyzed the central conduction time, defined as the time difference between ABR wave II and wave IV. There is no significant difference in the latency of ABR waves, indicating that peripheral and central conduction time are not altered in *Bdnf* cKO. We added the data for peripheral and central conduction to the manuscript (Figure 9).

3) There is no evaluation of whether loss of bdnf affects the density of oligodendrocyte progenitors or myelination (or Schwann cells). There are numerous reports that BDNF signaling through TrkB alters progenitor dynamics and the process of myelination. If myelination is altered, it could affect activity propagating along GBC axons and therefore maturation of their terminals. It is possible that there is sufficient BDNF coming from other sources to compensate, but if that is true, why don't these other sources compensate for the loss of oligodendrocyte-derived bdnf?

This manuscript addresses the role of OL-driven BDNF in synaptic transmission. We are studying the effect of BDNF on myelination in the auditory nervous system in another project. We examined axon myelination in BDNF cKO mice using electron microscopy to evaluate whether loss of OL BNDF impacts myelination, and found that there is no significant difference in axon diameter and myelin thickness by analyzing *g*-ratio.

Reviewer #2:[…] 1) The authors framed the entire story that BDNF released specifically from oligodendrocytes are critical for modulating the RRP in the context of the critical period of development. However, the choice of age group for different experiments is quite confusing. For example, patch-clamp recordings of EPSCs were from P16-20 mice (mature or nearly mature synapses); immunofluorescence labeling and EM were done in P20-25 mice; and calcium currents and capacitance measurements were done in P9-13 mice (immature synapses). Given that dramatic changes in presynaptic spike waveform, calcium channel-SV coupling distance and morphological remodeling all occur during this period, it is difficult to discern if oligodendrocyte derived BDNF is specifically important for developmental remodeling of immature synapses and/or synaptic signaling in mature synapses. It would impart readers with more confidence if the authors can show the results from the experiments in parallel age groups.

We appreciate the reviewer’s constructive comments. To address the concerns about the influence of age, we added new data for two different time points in control and *Bdnf* cKO. We compared the data from postsynaptic EPSC recordings, presynaptic capacitance measurements, and EM in the immature calyx synapse (at P10-P12), and we demonstrated EPSC recordings and EM at the mature calyx synapse (at P16-P20) in parallel. The data from both EPSC recordings and EM indicate a sustained effect of reduced OL BDNF at P16-P20. Taken together, OL-derived BDNF regulates presynaptic RRP in both the immature and mature synapses. We added new data in Figure 6 and Figure 6—figure supplement 1.

2) With the entirely opposite results from Baydyuk et al., 2015, it is essential to show and validate the expression patterns of BDNF/TrkB for P9-13 age group in order to strengthen the arguments made in this paper against the previous paper.

The conflicting findings may result from the differences between species, ages, or BDNF application method. In Baydyuk et al., 2015, the authors show BNDF and TrkB expression in calyces using BDNF and TrkB knock-in mouse strains, but BDNF and TrkB expression were not shown in WT animals. However, all electrophysiological experiments had been done in the brainstem of Wistar rats. Thus BDNF effects could be influenced by species or strain of experimental animals. Another possibility is the BDNF application method influences the results as acute exposure likely differs from in vivo knockout. In our current study, we detected TrkB signals from calyces in both control and cKO mice in the same tissue used for electrophysiology. BDNF expression was detected in OLs as well as MNTB neurons in control, whereas OL BDNF expression was significantly reduced in cKO mice. We added BDNF and TrkB immunostaining in the MNTB from control and cKO mice to the manuscript (Figure 5—figure supplement 2 and Figure 4).

3) In the capacitance jump experiments, it was stated that there is no difference in the basal membrane capacitance between WT and BDNF(+/-) calyces (Figure 3), but the calyx volume is significantly smaller in morphological analyses (Figure 6). There is an obvious conflict here which needs some explanation.

In both the *Bdnf*^+/-^ and cKO mice, we found no difference in the basal membrane capacitance in electrophysiological recordings (20 ± 1.5 pF, *n* = 10 for WT vs 21.5 ± 1.35 pF, *n* = 9 for *Bdnf*^+/^-, *P* = 0.43; Figure 3C, and 15 ± 1.9 pF, n=14 for control vs 15 ± 3.5 pF, n=17 for *Bdnf* cKO, Figure 5). In *Bdnf* cKO mice (at P10-12), we patched calyces and filled Alexa 568. After taking confocal images, we did volumetric 3D-reconstruction of calyx images and analyzed their volume using Amira software. We found that the volume has a tendency toward a slight reduction in the *Bdnf* cKO, but it was not significantly different from control. The volume of the calyx terminal was not significantly different in *Bdnf* cKO mice (1,378 ± 143.7 mm^3^, *n* = 7 for control and 1,199 ± 146 mm^3^, *n* = 6 for *Bdnf* cKO;*P* = 0.3308, Figure 7—figure supplement 1). We revised the manuscript and the figure to reflect this finding.

4) Given the abundant expression of TrkB in MNTB neurons, it is essential to analyze the effects of exogenous ligands on the amplitude and frequency of mEPSCs to ensure the observed effects is purely presynaptic as claimed (Figure 4). It is puzzling in fact that BDNF is not doing anything to postsynaptic neurons with much more abundant TrkB expression to detect BDNF than presynaptic terminals. It is probably too hasty to rule out the role of BDNF in postsynaptic signaling at this stage.

It is well known that BDNF regulates neuronal survival and growth in the developing brain. Thus BDNF could be involved in postsynaptic modulation and cellular signaling. In this study, we focus on studying the role of oligodendroglial BDNF in presynaptic properties and neurotransmitter release. In our exclusive experimental design using the *Bdnf*^+/-^ or cKO mice, we did not observe postsynaptic alterations although we could not rule out the role of BDNF in postsynaptic signaling.

Reviewer #3:[…] 1) The mechanism by which BDNF rescues the RRP is unclear. The authors clearly show that with their capacitance measurements there is a reduction in the readily releasable pool at the calyx and that this can be rescued by application of BDNF. In addition, in the absence of BDNF their EM data clearly demonstrates a reduction in docked SV. Therefore, two possible interpretations can be made. Either each calyx contains the same number of active zones (AZs), but in the absence of each AZ has less docked SVs or absence of BDNF during development results in a reduction in the number of AZs. In both cases application of BDNF would result in an increase of docked SVs which would result in an increase on the RRP. The authors should show with EM that BDNF rescues the docking defect seen. In addition, the authors should carry out nonstationary EPSC variance analysis to determine the number of functional AZs in the presence and absence of BDNF.

As it was mentioned above, we added new data for the rescue of the docking deficit using EM, which show that the TrkB agonist (7,8-DHF) rescues the deficit in docked vesicles at the calyx terminal shown in cKO mice (Figure 7). We did not include the number of AZs in this manuscript, and mentioned the possibility of changes in the number of AZs in the Discussion.

2) The data concerning BDNF effects on the voltage dependence of activation on Calcium channels is not convincing. These measurements are highly dependent on the pipette series resistance in whole cell voltage clamp. There is no mention of the pipette series resistance during the actual recordings. Based on the example traces presented there appears to be differences in the tail current decay between WT and BDNF^+/-^. This could be due to differences in the quality of the voltage clamp or that reductions in BDNF do impact tail currents. In addition, it is not known if a leak subtraction protocol was performed. The authors should redo their IV analysis with better lower residual patch clamp pipette resistance. Furthermore, they should perform the IV analysis in 1.0 mM external Ca^2+^ to also offset potential issues with voltage clamp quality.

In the revision, we re-examined the voltage-dependence of Ca^2+^ channel currents in the calyx terminals from WT, *Bdnf*^+/-^, control, and *Bdnf* cKO in whole-cell voltage clamp with a lower series resistance and re-analyzed them after leak subtraction using the “traditional” p/4 stimulus train in EPC10-Patchmaster. We addressed the series resistance during the recordings in the Materials and methods; the series resistance was 15 ± 3.1 MOhm before compensation (n=10), adjusted to <10 MOhm after compensation of ~ 50%. When the patch recordings maintain a lower series resistance to avoid potential issues with voltage clamp quality, there is no difference in the voltage dependence of Ca^2+^ channels between WT and *Bdnf*^+/-^ mice. The *I*-V curve was newly updated in Figure 3. We added the newly analyzed data and representative traces in Figures 3 and 5.

3) The authors demonstrate a reduction in AP evoked release and RRP as defined trains of APs at 100 Hz. It has been well established at the age groups p9-11 that a 3ms step depolarization corresponds to the RRP that can be released by action potentials. In Figure 5 which utilized p9-13, there appears to be no difference between the amount of exocytosis with a 2ms step pulse. Therefore, it is unclear why there is a dramatic effect in the AP RRP measurement and RRP as measured by the step depolarizations. It would be helpful if the authors did a 3ms Cm measurement to directly compare. Additionally, the authors should compare data collected at P9-11 vs P12-13 as mice begin to hear at P12. To accurate determine effects on BDNF regulation of the calyx RRP paired recordings at P9-11 would be ideal but not critical to their interpretations. Finally, why did they not carry out fiber stimulation at this age group too?

We added data with the 3 ms step pulse at P9-12 in Figure 5. In control, 2- and 3-ms depolarizing pulses induced a capacitance jump (ΔC_m_) of 51 ± 8.6 fF and 72 ± 16.5 fF, n=12, respectively, which are comparable to those in age-matched mouse calyces, previously describe in Lin and Taschenberger, 2011. In *Bdnf* cKO calyces, ΔC_m_ in response to 2-ms depolarization was difficult to resolve, and 3-ms depolarization induced a ΔC_m_ of 27 ± 10.5 fF, n=10. A longer depolarization induced a larger ΔC_m_, and pulse durations > 40 ms exhibited saturation of ΔC_m_ in both control and cKO calyces (Figure 5E, Lin et al., 2011).

For the fiber stimulation, we added new data for evoked EPSCs in immature (P10-12, before hearing onset) and mature (P16-20, after hearing onset, Figure 6) MNTB neurons from control and cKO mice.

4) The calyx of Held goes through structural changes after hearing onset. The volumetric reconstruction of the P20-25 age group revealed significant reductions in calyx volume. However, the Capacitance measurements revealed no change in calyx volume as measured as pF. However, this comes from a different age group compared to the group in volume reconstructions. I am more inclined to trust the volumetric reconstructions since it is very difficult to accurately determine the calyx size with the patch clamp experiments. The authors should carry out volumetric reconstructions as the same age group or measure the calyx volume via a capacitance measurement in the P20-25 age group. Otherwise it is difficult to know why there is such a difference between the two measurements of calyx volume.

Thanks for the reviewer’s thoughtful comments. The volumetric reconstruction from P10-12 calyces after presynaptic capacitance measurement was reanalyzed. There was no difference in the membrane capacitance and the volume of the calyx from 3D reconstruction between control and cKO (at P10-12). The volume of the calyx terminal was not significantly different in *Bdnf* cKO mice (1,378 ± 143.7 mm^3^, *n* = 7 for control and 1,199 ± 146 mm^3^, *n* = 6 for *Bdnf* cKO;*P* = 0.3308, Figure 7—figure supplement 1). This result was comparable to their membrane capacitance (C_m_) measurement; there was no difference (15 ± 1.9 pF, n=14 for control vs. 15 ± 3.5 pF, n=17 for *Bdnf* cKO, Figure 5). We added this new data in Figure 7—figure supplement 1.

**Author response image 1. respfig1:** (Left) 3D images of calyces from control and cKO mice. (Right) Summary of volume and Cm.

[Editors' note: the author responses to the re-review follow.]

Reviewer #1:Jang and colleagues have made substantial revisions to the manuscript and performed some very nice additional experiments to address concerns raised in the prior review. However, the additional data provided to address concerns about the specificity of recombination in CNPase mice are not sufficient.1) In Figure 5—figure supplement 1, the authors show immunostaining for NeuN, GFAP, Pdgfra, CC1 and GFP to assess which cell types exhibit recombination. In these panels, GFP refers to GCaMP6f in CNPase-Cre x R26-lsl-GCaMP6f mice. There is diffuse GFP immunoreactivity throughout the GFP panels that is unaccounted for and unexpected if expression were restricted to oligodendrocytes. In addition, these images reveal intense regions of GFP immunoreactivity that co-localize with NeuN (cell in the center left of panel A) and co-localize with Pdgfra (panel C). Perhaps most significantly, in the GFP image of panel C there are several long processes and one cell body (center left) that is much larger than one would expect for an oligodendrocyte. For some reason, although this cell looks like a neuron, it also exhibits weak immunoreactivity to CC1. However, the CC1 immunoreactivity in panel C is very unusual, with several long processes visible (this is not normally observed with CC1 – see panel B) and many of the CC1+ cell bodies have unusual morphologies. Since orthogonal projections are not shown in this figure, it is difficult to assess whether labeling patterns show proximity or true co-localization. Thus, these immunostaining data to not provide strong support for the conclusion that CNPase-Cre mice exhibit selective recombination in oligodendrocytes. A minor point, in the figure legend, the mice are referred to as "CNPase-GCaMP6f-GFP mice". Please indicate the exact genotype of the mice for clarity.

To address the concerns about the immunohistochemical analysis we have added a new representative image with orthogonal views orthogonal views to aid in the visualization of the colocalized cells. In addition we added quantitative analysis of co-localization using two different neuronal markers (NeuN and Map2) and two reporter lines (GCaMP6f and tdTomato reporter lines). The quantitative analysis demonstrates that >97% of the recombined cells are OLs with a very small portion of neurons showing positive staining (< 3%, See *the Figure 5—figure supplement 1*).

2) The authors also perform PCR analysis on isolated cells to determine the specificity of recombination. However, no experimental details are provided for the FACS and PCR analysis shown in Figure 5. There is insufficient information provided that the isolation procedure was able to isolate a representative population of cells from the brainstem (astrocytes, neurons, oligos, etc.). Procedures used to preserve glia often result in widespread death of neurons, so it is important to assess what types of cells are in this pool.

We have added additional qPCR analysis to include the neuronal marker, Kcc2. There is no reduction of the Kcc2 in sorted neuronal populations. Experimental details have been added to the Materials and methods section. We carefully verified the isolated cells. Isolated cells from *Cnp^cre+/-^*; GCAMP6f-GFP+ cells versus GFP- cells demonstrated an enrichment of Olig2 RNA quantified by qPCR. Kcc2, was depleted from the GFP+ population, indicating that GFP+ cells are Olig2+ and Kcc2 negative OLs (see *the Figure 5—figure supplement 2*).

3) The criteria used to determine positive versus negative O1 cells in Figure 5—figure supplement 2A is not indicated. From the second panel in A, it looks like there is a clear break between the blue/magenta cells and the green cells. However only the O1+ cells that exhibited the highest immunoreactivity were selected for the PCR analysis. This analysis should be repeated by including the blue cells, to provide a more representative sample of the population. It is unclear why GFP was not used to isolate cells in this example, as this would provide a more direct assessment of what types of cells experienced cre-dependent recombination. Because there is concern about neuronal recombination, neurons from this region should be isolated and GFP and BDNF expression assessed selectively in this pool.

The selection of the positive population in FACS was performed using scatter, histogram analysis, and the expertise of the FACS CORE. We have added data from *Cnp^cre^*: R26-lsl-GCaMP6f mice that demonstrate an enrichment of Olig2 in the GFP+ cells and an absence of Kcc2. We have not yet crossed these mice with the *Bdnf ^flox/flox^*, thus do not provide data from those mice.

4) The authors indicate that >27 cycles were necessary to detect BDNF message in the O1+ population. 27 cycles is typically used as a cutoff, with any product observed with more cycles being attributed to noise/non-specific amplification. This suggests that the amount of BDNF in O1+ cells is at the limit, or below the limit, of detection. Although more cycles were required to detect a product in the cKO O1+ cells, confidence in these data are low for this reason.

We demonstrate that the *Bdnf* primers detect a reduction in *Bdnf* RNA and thus are specific using *Bdnf* heterozygous mice compared to littermate controls. We show that the *Bdnf* primers linearly amplify even with low amounts of input RNA and cycles >27, indicating that the *Bdnf* primers can detect differences in RNA within the CT range observed in these studies (see the Materials and methods section).

5) In Figure 5—figure supplement 2D immunostaining is performed in the cKO mice. The panels show that BDNF immunoreactivity is somewhat lower in O1+ cells in the cKO mice. As a constitutive cre was used, it is surprising that BDNF expression was not abolished. It also appears that BDNF immunoreactivity is also lower in neurons in the cKO mice. Thus, these data raise further questions about the specificity of recombination in these mice.

Due to concerns about the specificity of the BDNF antibody, we have removed the immunostaining. As we have shown with immunohistochemistry, the *Cnp^cre^* targets a subpopulation of OLs. Thus, the presence of BDNF in the O1+ from *Bdnf* cKO mice likely comes from *Cnp^cre^*^-/-^ O1+ OLs.

Reviewer #2:1) In general, for all bar graphs it would be ideal if the authors also provided a dot plot on top of the bar graphs. This would allow the reader to get an idea of the distribution of the date for the experiments.

We did not change the bar graphs to the dot plot on the bar graphs, saving our efforts on the remaking all figures. Instead, we focused on adding more data to complete the manuscript following reviewer’s comments. All data set will be shared, thus if anyone wants to look the distribution of the data, it will be available to access the data set.

2) In Figure 2 it would be ideal to also report the release probability measurements as determined by EQ plot measurement. Please report release probability measurements for EQ and SMN in Figure 6.We did. The release probability measurements as determined by EQ plot measurement was added in the main text (Figure 2). Release probability measurements for EQ and SMN were reported in Figure 6.3) Figure 3 graphs appear to be missing the 3ms data point on all Panel B and Panel D. There are 5 measurements corresponding to the different depolarization times but only 4 data points on these graphs. In addition, in Panel F of the IV data it appears there is a shift in voltage dependence of activation. There is a lot of variance at the -40 mV point but the mean looks different. The authors should make a simple statement to address this potential issue as it could be due to differences in voltage clamp.We added the missing points for 3 ms in Figure 3B and D. We added more data for the IV curve in Figure 3F, showing no difference the mean value at the -40 mV.4) Figure 5 appears to be missing plotted IV data. In fact, I am confused by these traces. In the figure legend, it states traces showing 100 ms depolarizations from -80 mV to 0 mV. But there is no panel lettering to these traces.We added the IV curve of Ca^2+^ current from control and cKO in Figure 5. The figure legend was revised.5) Figure 7 is missing representative EM images from the immature calyx synapse. They need to be added.We added representative EM image from the immature calyx synapse to Figure 7—figure supplement 1.6) Figure 7—figure supplement 1 is missing. Also, the volumetric data on the P20-25 calyces that was included in the previous version of the manuscript is now missing.We added Figure 7—figure supplement 2, including the volumetric data.6) Figure 9. It would be helpful to delineate to the reader who are not auditory physiologist what the ABR waves correspond to.

We did.

Reviewer #3:This revised version has significantly improved in its quality. The authors have addressed virtually all my criticisms. The action potential waveform is very important for the argument that RRP is specifically altered by BDNF from oligodendrocytes. I would suggest that stats for spike waveform in two age groups are included in the main figure or text.

We added the statistic data for the amplitude and half-width of spike waveform from WT and hetero calyces to the Results section.

[Editors' note: further revisions were requested prior to acceptance, as described below.]

Thank you for resubmitting your work entitled "Oligodendrocytes regulate presynaptic properties and neurotransmission through BDNF signalling" for further consideration at eLife. Your revised article has been favorably evaluated by Gary Westbrook as the Senior Editor, and Dwight Bergles as the Reviewing Editor. The manuscript has been improved, but there are some remaining issues that need to be addressed before acceptance, as outlined below:The final conclusion of the study is that oligodendrocytes in the brainstem release BDNF during development, which regulates the structural and functional maturation of calyceal synapses in the mouse MNTB. The effects on glutamate release are profound and well described through paired recordings and anatomical studies. The primary limitation of the study is that one mouse line, Cnp-Cre, was used to alter BDNF levels in oligodendrocytes. Because there is concern about the specificity of recombination in this mouse line, the study would be stronger if a subset of the effects were also assessed in a second oligodendrocyte specific Cre line.

We recently established a second *Bdnf* cKO mouse (*Sox10-*creER: *Bdnf*^f/f^) using a second oligodendrocyte-specific Cre line (Sox10-creER mice, provided by Dr. Shin Kang, Temple University). In this second cKO mouse, we found the same phenotype of presynaptic terminals in the calyx of Held synapse with those described in the manuscript (see Author response image 2). We are preparing a separate manuscript using this mouse line, thus the following data set is not included in the current manuscript.

**Author response image 2. respfig2:** Effects of reduced BDNF in OL on vesicular glutamate release from presynaptic terminal in the *Bdnf* cKO (Sox10creER: *Bdnf*^f/f^). (A) Representative traces for membrane capacitance (Cm; top) and Ca^2+^ current (*I*Ca; bottom) induced by 10‐ms depolarization from P11–12 calyx terminals in control (black) and *Bdnf* cKO (red) mice. (B‐C) Summary of Ica and Cm induced by 10‐ms depolarization to 0 mV in calyces from control (black) and cKO (red) mice. (D) Depolarization duration plotted against ΔCm for control and *Bdnf* cKO mice. (E) Verification of Cre-line specificity. Using FACS and qPCR, isolated GCaMP6f‐GFP+ cells demonstrated a high level of Olig2 mRNA, a oligodendrocytes marker, but no detectable Kcc2 mRNA, a neuronal marker, indicating this Cre‐line is specific to Olig2+ oligodendrocytes.

The authors did not include a reporter in the Cnp-Cre x Bdnf^flox/flox^ cross to allow assessment of BDNF levels in the recombined cells and the selectivity of the recombination, instead relying on the surface antigen O1 for isolation of oligodendrocyte lineage cells for subsequent qPCR analysis. The authors show that samples enriched in oligodendrocyte lineage cells (O1+) have detectable bdnf mRNA by qPCR and that this level is reduced in the cKO mice. Although there is a trend towards lower bdnf mRNA in the O1- fraction, the value was not significantly lower (n=3). However, they did not specifically measure whether bdnf was lower in neurons (the O1- population presumably contains other cells, such as astrocytes which could mask any effects on neuron bdnf levels, as levels are typically higher in those cells).

We added new data for the assessment of *Bdnf* mRNA levels in the recombined cells and selectivity of the recombination using the *Cnp^cre^*:Bdnf ^f/f^;GCaMP6f-GFP mice (n=3 animals). The result indicated 1) the presence of *Bdnf* mRNA in OLs in the MNTB of the brainstem from control, and 2) the knock-down of *Bdnf* mRNA expression in OLs in *Bdnf* cKO mice (Figure 5—figure supplement 2).

In addition, there is no quantification of BDNF protein in oligodendrocytes at this age, and no immuno-localization or in situ hybridization indicating the presence of BDNF protein or bdnf mRNA message in oligodendrocytes in this region. The authors find a small proportion of neurons in the brainstem exhibit evidence of recombination in this mouse line, and recombination outside the oligodendrocyte lineage (e.g. Schwann cells, immune cells) in Cnp-Cre mice has been described by others. This is significant, because BDNF levels are much higher in neurons than oligodendrocytes in cortical regions, although to our knowledge this hasn't been explored in the brainstem at this developmental stage. As BDNF is a secreted protein, any reduction in other cells could therefore contribute to the effects observed.

Our results showed the presence of BDNF in oligodendrocytes in the MNTB of the mouse brainstem during early postnatal development. ~ 50% of Olig1+ oligodendrocytes were positive for BDNF immunostaining (Figure 1A). Using FACS and qPCR, we demonstrated that O1+ cells have detectable *Bdnf* mRNA, which was significantly reduced in *Bdnf* cKO mice. In addition, GCaMP6f-GFP+ cells also showed a substantial amount of *Bdnf* mRNA in control, which was significantly reduced in *Bdnf* cKO mice (newly added to Figure 5—figure supplement 2). There was no significant difference in the level of *Bdnf* mRNA in the O1− fraction or GFP- fraction in cKO versus control, although there was a trend towards lower *Bdnf* mRNA. The sorting likely did not completely isolate the oligodendrocytes and the remaining population could contribute to the slight reduction in *Bdnf* mRNA. Another possibility is the small percentage of neurons that are affected is contributing to this trend. However, it is unlikely that BDNF reduction in <5% neurons have a global and major impact on the synaptic function in the MNTB. Due to the biochemical nature of BDNF, it is thought to act locally at the synapse with limited diffusion (Sasi, 2017). The effect of BDNF within a synapse has been observed to occur with the low micrometer range (within a distance of 4.5 µm, Horch, 2002). Thus, BDNF reduction in the small portion of neurons (<5%) may not critically contribute to the synaptic phenotype observed in the cKO. We interpret that functional alterations of the calyx synapse were caused by the loss of BDNF in oligodendrocytes, which constitute the majority of CNP-expressing cells. As we addressed above, studies using additional oligodendroglia cre line such as Sox10-CreER mice confirm this interpretation.

At a minimum, the authors need to thoroughly address these limitations in the Discussion.

We rewrote the Discussion including the issue about the Cre line raised by the reviewer and editor.

Additional concerns:1) All data points should be added to the histograms, so that the readers can assess the distribution of values from individual experiments.

We changed all bar graphs to include all data points.

2) It would be helpful if the authors determined the identity of the tdTomato+ cells in Figure 5—figure supplement 1G by immunostaining.

We added the panel showing the identity of the tdTomato+ cells, which are positive for Olig2.

3) There are numerous typographical errors that need to be corrected (e.g. Figure 5—figure supplement 1 – "Nuen"), the labels in some panels are unclear, and some formatting is off (e.g. Figure 5—figure supplement 2D). Throughout the study the authors used GCaMP6F as a reporter to show the specificity of recombination in the CNP-Cre line, but indicate "GFP" or GCaMP6f-GFP in the graphs and panels. This needs to be corrected throughout, for example, by indicating anti-GFP when indicating the antibody (rather than the protein) or GCaMP6F when indicating the protein.

We corrected throughout.

4) The qPCR primer design should be changed so that the primers span across exons to reduce the possibility of genomic signal contamination.

As instructed in the RNAqueous Micro Kit (Invitrogen AM1931), DNA is removed by enzymatic digestion. To ensure that the primer used was not detecting genomic signal contamination and was specific to cDNA, the qPCR was run using isolated RNA (with prior DNA removal or not) without cDNA synthesis, and this control had no amplification. Therefore, we conclude that the primer is specifically amplifying cDNA and does not amplify DNA or RNA prior to cDNA synthesis.

5) The species should be indicated in the title.

We revised the title to include the species.